# SpikeVLA: Vision-Language-Action Models with Spiking Neural Networks

**Ruiqi Song** [* 1 2 3]  **Dujun Nie** [* 2 3 4]  **Siyu Teng** [3 5]  **Baiyong Ding** [2 6]  **Xiaotong Zhang** [2 3 4]  **Dong Li** [3 7]
**Chenming Zhang** [3 6 8]  **Yuchen Li** [3 9]  **Hangbin Wu** [1]  **Long Chen** [2 3 6 8]

## Abstract

Vision-Language-Action (VLA) models have become a dominant paradigm for embodied intelligence. However, most existing approaches are built on large-scale transformers, resulting in substantial inference latency and energy consumption that limit their practical deployment in low-power, real-time scenarios. We propose SpikeVLA, a spiking VLA architecture for embodied navigation with energy-efficient inference, consisting of three key components. (i) A spiking vision encoder, Spike-V, that replaces dense continuous layers with event-driven spiking layers to reduce the energy consumption of visual representation learning. (ii) A multi-modal spiking large language model, Spike-L, that reformulates cross-modal reasoning with spiking dynamics and token-level event-driven sparsity to further lower computational cost. (iii) A spiking action policy network, Spike-A employs Laplacian-kernel population coding with a multi-layer fully connected SNN, and decodes spiking activities into stable and robust continuous control with energy-efficient inference under low-power constraints. Experiments on navigation and robotic control tasks show that SpikeVLA significantly reduces energy consumption and computational cost while maintaining competitive performance, highlighting its potential for low-power, real-time embodied intelligence.

---

[*]Equal contribution [1]College of Surveying and Geo-informatics, Tongji University, Shanghai, China [2]State Key Laboratory of Multimodal Artificial Intelligence Systems, Institute of Automation, Chinese Academy of Sciences, Beijing, China [3]OpenSpace Lab, China [4]School of Artificial Intelligence, University of Chinese Academy of Sciences, Beijing, China [5]Shenzhen University, Shenzhen, China [6]Waytous, Wuhan, China [7]Faculty of Innovation Engineering, Macau University of Science and Technology, Macau, China [8]IAIR, Xi'an Jiaotong University, Xi'an, China [9]Department of Informatics, Technical University of Munich, Munich, Germany. Correspondence to: Hangbin Wu <hb@tongji.edu.cn>, Long Chen <long.chen@ia.ac.cn>.

*Proceedings of the 43rd International Conference on Machine Learning*, Seoul, South Korea. PMLR 306, 2026. Copyright 2026 by the author(s).

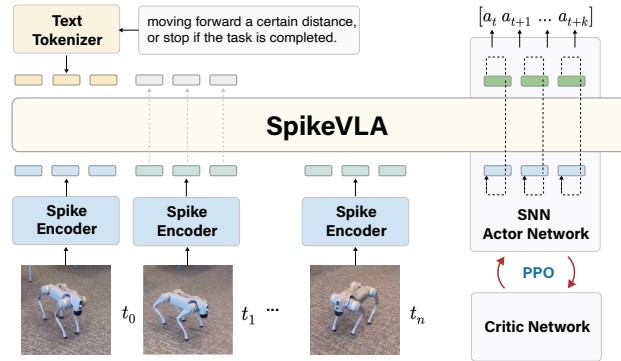

Figure 1: SpikeVLA: Vision-Language-Action Models with Spiking Neural Networks.

## 1. Introduction

In recent years, Vision-Language-Action (VLA) models have become increasingly prominent as a key paradigm for embodied intelligence, unifying vision, language, and action to support navigation, manipulation, and collaboration tasks in complex environments. Despite significant progress in embodied intelligence, the rising inference-time compute footprint and energy demands of large VLA models increasingly constrain on-board deployment on resource-limited platforms (e.g., micro-robots, legged systems, and deep space exploration robots) and latency-critical real-time operation, which is particularly challenging for space exploration robotics. Therefore, lightweight and energy-efficient VLA models are critical for the scalable real-world deployment of embodied intelligence.

Recent VLA models such as NaVid (Zhang et al., 2024b), NaVILA (Cheng et al., 2025), and UniNaVid (Zhang et al., 2024a) typically rely on large-scale transformers for multimodal reasoning, together with ANN-based policy networks for control. Although they perform strongly on instruction-driven navigation, their dense computation imposes considerable energy and latency costs, limiting deployment on resource-constrained platforms and in latency-critical scenarios. To address this challenge, previous work has explored efficiency-oriented designs. For example, COSMO (Zhang et al., 2025b) reduces inference cost

through selective memorization, while VL-Nav (Du et al., 2025) focuses on efficient spatial reasoning with real-time deployment. Nevertheless, these methods are still grounded in continuous and dense computation, which offers limited headroom for further efficiency improvements. In contrast, event-driven spiking neural networks introduce a fundamentally different, energy-aware paradigm for computation. They adopt an event-driven sparse spiking mechanism that triggers computation only when information changes, significantly reducing redundant operations and energy consumption, which introduces a new paradigm for low-power VLA deployment.

To address these challenges, we propose SpikeVLA, the first VLA architecture built on spiking neural networks, which represents a trade-off between performance and efficiency, as shown in Fig. 1. SpikeVLA consists of three complementary modules. Spike-V provides energy-efficient visual representations through event-driven spiking visual encoding. Spike-L reformulates multimodal reasoning with spiking dynamics and token-level sparsity to reduce inference-time compute and energy cost. Spike-A achieves stable and robust continuous control with a fully spiking action policy, supporting low-energy operation. Together, these components establish SpikeVLA as a neuromorphic computing paradigm for embodied intelligence.

In summary, this work makes three main contributions. **1)** We propose SpikeVLA, the first VLA framework built on spiking neural networks. Through event-driven sparse activations, SpikeVLA substantially reduces energy consumption in reasoning and control, shifting VLA from dense continuous computation toward a low-power, real-time neuromorphic paradigm. **2)** We develop a three-module spiking architecture: Spike-V for event-driven spiking visual encoding, Spike-L for energy-efficient multimodal reasoning with token-level sparsity, and Spike-A for stable continuous control with a fully spiking policy network. **3)** We conduct extensive experiments on Vision-Language Navigation and robotic control tasks, demonstrating that SpikeVLA matches the performance of ANN-based baselines while reducing energy consumption and computational cost, and serves as a general energy-efficient design paradigm transferable to other robotic VLA tasks.

**Conflict of Interest Disclosure.** The authors declare that they have no financial conflicts of interest related to this work. All research and writing were conducted independently and solely for academic purposes.

## 2. Related Work

### 2.1. Vision-Language-Action for Navigation

Early vision-and-language navigation methods in continuous environments relied on end-to-end action prediction,

as represented by Seq2Seq (Krantz et al., 2020). However, these methods often suffer from error accumulation and semantic drift in long-horizon rollouts. Some approaches use waypoint-based and hierarchical planning to decouple high-level intent from low-level execution (Krantz et al., 2021; Raychaudhuri et al., 2021; Chen et al., 2021). Moreover, subsequent research has increasingly focused on the generalization of the cross-scene and the integration of spatial priors (Hong et al., 2022a;b; Krantz & Lee, 2022). In parallel, methods such as CM2 (Georgakis et al., 2022) and WS-MGMap (Chen et al., 2022) have been proposed. For longer horizons and more complex environments, recent approaches have increasingly focused on global state maintenance, memory, and online closed-loop operation (Wang et al., 2023c; Hong et al., 2023; Wang et al., 2023a;b). Recent research has increasingly focused on enhanced planning, representations, unified modeling, and streaming closed-loop navigation (An et al., 2024; Wang et al., 2024; Zhang et al., 2024b;a; Long et al., 2024; Wei et al., 2025; Yu et al., 2025). However, this trend also increases the computational cost of continuous inference during long-horizon closed-loop operations. As a result, the demand for low-power, real-time VLA deployment is increasing, driving the need for efficiency-oriented designs (Zhang et al., 2025b; Du et al., 2025). However, these methods primarily operate within a continuous, dense-computation paradigm, which limits further efficiency gains and highlights the need for neuromorphic, event-driven VLA models like SpikeVLA.

### 2.2. Spiking Neural Networks

Spiking neural networks (SNNs) are promising for efficient temporal learning, thanks to their event-driven computation and potential for energy savings. Many works use ANN-to-SNN conversion to preserve ANN capabilities while facilitating low-power inference, including (Oh & Lee, 2024; Bu et al., 2025; Huang et al., 2024; 2025). In parallel, efficient spiking architectures have been developed to improve vision performance and provide general building blocks for multimodal and long-horizon scenarios (Lei et al., 2025; Zhang et al., 2025c). Building on these foundations, recent works have extended spiking models to foundation models through generative pretraining, distillation, enhanced spiking mechanisms, and more efficient spike allocation (Bal & Sengupta, 2024; Xing et al., 2024b;a). More recently, SNNs have been increasingly explored in embodied domains such as continuous control. Previous works have enhanced the expressivity of spiking policies through fully spiking actor designs, while also addressing the mismatch between discrete spiking dynamics, such as Proxy Target (Zanatta et al., 2024; Zhang et al., 2022a; Xu et al., 2025). Applications in autonomous driving further demonstrate the feasibility of SNNs for long-horizon tasks under the efficiency constraints platform (Zhu et al., 2024). Inspired by these developments,

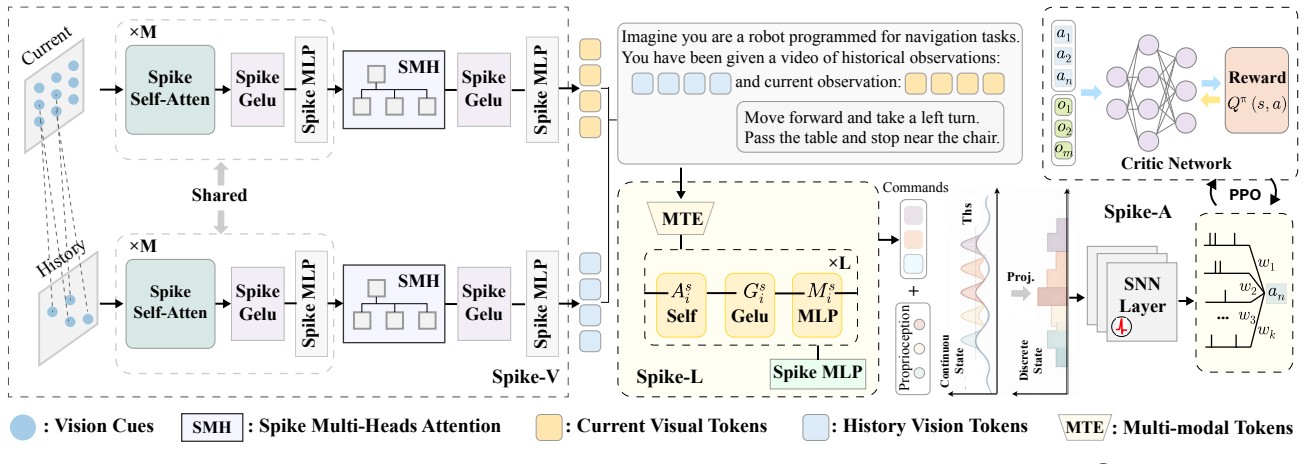

Figure 2: **Architecture of SpikeVLA.** We introduce an SNN-based VLA architecture composed of a spiking neural network vision encoder, a multimodal spiking large language model, and an SNN action policy. Under a unified event-driven paradigm, this design establishes a new architectural paradigm for VLA.

SpikeVLA is designed for energy-constrained deployment, using an event-driven spiking architecture to reduce inference costs.

## 3. Method

### 3.1. Architecture

We propose SpikeVLA, an end-to-end spiking VLA architecture for embodied navigation, consisting of a spiking vision encoder (Spike-V), a multimodal spiking language model (Spike-L), and a fully spiking action policy network (Spike-A), as illustrated in Fig. 2. Spike-V takes the current frame along with a sequence of past frames, encoding temporal observations into event-driven spiking visual tokens for energy-efficient visual encoding (see Section 3.2). Spike-L fuses visual and text tokens and performs event-driven channel-wise sparsification to reduce computational cost (see Section 3.3). Spike-A maps the fused multimodal representation to continuous actions with a fully spiking policy network, achieving stable and robust closed-loop control under low-power constraints (see Section 3.4).

### 3.2. Spike Neural Network Vision Encoder

We propose an SNN-based visual encoder that fuses the current frame with history frames to provide temporal context for time-dependent VLA tasks. Built upon a spiking Transformer, it replaces dense continuous feature extraction with sparse, spike-driven computation over discrete timesteps. We reformulate linear layers and key nonlinear operations in spiking form, reducing computation and energy while preserving representational capacity for efficient temporal visual feature extraction.

**Differential Spiking Neurons.** To preserve high-fidelity representations with few timesteps $T$, we adopt differential coding to express continuous activations as incrementally accumulated updates over time, so that activation updates in both linear mappings and nonlinear transformations follow a unified incremental recursion. This process can be written as follows:

$$\delta^l[t] \;=\; \bar{a}^l[t-1] \;+\; \theta^l \, z^l[t]$$

$$\bar{a}^l[t] \;=\; \bar{a}^l[t-1] \;+\; \frac{\theta^l}{t} \, z^l[t] \;=\; \frac{1}{t}\sum_{i=1}^{t}\delta^l[i], \tag{1}$$

where $l$ denotes the layer index and $\theta^l$ is the firing threshold of spiking neurons in layer $l$. $z^l[t] \in \{0,1\}$ indicates whether a spike is emitted at time step $t$. $\delta^l[t]$ represents the encoded output produced by differential coding, while $\bar{a}^l[t]$ is the average of $\delta^l[t]$ from time 1 to time-step $t$ with initialization $\bar{a}^l[0] = 0$.

Building on differential coding, we instantiate spiking units as differential spiking neurons, supporting stable spiking dynamics and sparse event-driven transmission with only a few timesteps $T$. We introduce a recurrent auxiliary state into membrane updates to incrementally correct the input current based on both input changes and emitted spikes, aligning neuron dynamics with the differential-coding recursion and promoting sparse event-driven transmission. The mechanism is formulated as follows:

$$I^l[t] = c_r^l[t] + x^{l-1}[t],$$

$$c_r^l[t+1] = c_r^l[t] + \frac{x^{l-1}[t]}{t} - \frac{\theta^l z^l[t]}{t}, \tag{2}$$

where $I^l[t]$ is the input current of the differential spiking neuron at time step $t$, and $x^{l-1}[t]$ is the output of the previ-

ous layer. $c_r^l[t]$ denotes the recurrent residual state for the input-current correction. $\theta^l$ is the firing threshold of the layer $l$, and $z^l[t] \in \{0, 1\}$ indicates whether a spike is emitted at the time step $t$, producing the output $x^l[t] = \theta^l z^l[t]$. Fully connected, convolutional, and matrix multiplication layers constitute the main computational bottlenecks, and inserting spiking neuron layers before them supports event-driven processing and reduces energy consumption.

**Linear-Layer Conversion.** We model compute-intensive mapping operations as linear layers and drive their computation with time-step increments generated by differential spiking neurons, resulting in event-driven execution.

Each neuron applies a linear mapping to its input at every time step. Equation (3) specifies the membrane-potential update for neurons in the $l$-th layer. To prevent bias accumulation, the bias term is moved into the initial membrane potential of the next unit. It is expressed as follows:

$$x^l[t] = W^l x^{l-1}[t], \tag{3}$$

Where $x^l[t]$ is the output of the $l$-th layer at time step $t$, $W^l$ is the weight matrix for the $l$-th layer, and $x^{l-1}[t]$ is the input of the $(l-1)$-th layer at time step $t$.

We build our visual encoder on SigLIPv2 and apply the proposed operation to linear layers, such as MLP blocks and attention projection matrices. This converts matrix multiplications and convolutions from dense continuous computation to event-driven operations.

**Nonlinear-Layer Conversion.** Beyond linear mappings, Transformers further require nonlinear operators such as normalization, activations, and attention computations that combine normalization with multiplicative interactions. To integrate these operators into time-step inference while preserving their functional forms, we employ differential graded units. The key idea is to output the incremental change at each time step, avoiding redundant recomputation and transmission of the full continuous value. This enhances fidelity under small $T$ conditions while not requiring additional fitting or retraining.

For any single-input nonlinear operator $F(\cdot)$, we realize its dynamics with a differential graded unit under differential coding during the ANN-to-SNN conversion. This mapping is given by the following equations, which convert continuous nonlinear operators into graded updates to improve efficiency and avoid redundant computation.

$$c^l[t] = c^l[t-1] + \frac{x^{l-1}[t]}{t},$$
$$\Delta_F^l[t] = F^l(c^l[t]) - F^l(c^l[t-1]), \quad x^l[t] = t * \Delta_F^l[t]. \tag{4}$$

where $F^l(\cdot)$ denotes the nonlinear operator in layer $l$ that takes a single input $x^{l-1}[t]$, and $c^l[t]$ denotes the membrane potential at time step $t$.

During ANN-to-SNN conversion, differential coding provides a dynamic realization for operators with two inputs. We maintain two states, $c_a[t]$ and $c_b[t]$, and define the mapping as follows:

$$c_a^l[t] = c_a^l[t-1] + \frac{x_a^{l-1}[t]}{t},$$
$$c_b^l[t] = c_b^l[t-1] + \frac{x_b^{l-1}[t]}{t},$$
$$x^l[t] = t * \left( c_a^l[t] \odot c_b^l[t] - c_a^l[t-1] \odot c_b^l[t-1] \right)$$
$$= x_a^{l-1}[t] \odot c_b^l[t] + x_b^{l-1}[t] \odot c_a^l[t] + \frac{x_a^{l-1}[t] \odot x_b^{l-1}[t]}{t}, \tag{5}$$

Where $c_a^l[t]$ and $c_b^l[t]$ represent the membrane potentials at time step $t$ for the two states, and $x_a^{l-1}[t]$ and $x_b^{l-1}[t]$ are the incremental inputs for these states at time step $t$. The symbol $\odot$ denotes element-wise multiplication. The output $x^l[t]$ is the result of combining the two inputs over time and serves as the updated value at the time step $t$.

We employ this mechanism to realize nonlinear operators that are not directly amenable to spiking conversion, including `LayerNorm`, `GELU`, and `Softmax` in attention, as well as multiplicative and matrix-multiplication composites within attention blocks. Meanwhile, the linear projections in attention are handled by the linear-layer conversion operation described earlier.

Building on differential coding, we introduce differential spiking neurons and perform unified differential conversion of linear and nonlinear operators in SigLIPv2, thereby obtaining a spiking vision encoder, SpikeSigLIP. It supports event-driven computation, improving computational efficiency and reducing redundant operations, while maintaining strong performance.

### 3.3. Multimodal Spiking Large Language Model

Building on LLaMA-8B, we perform supervised fine-tuning by integrating real video, simulated data, auxiliary navigation, and VQA datasets to construct a foundational model for embodied navigation. Building on this, we integrate spike encoding mechanisms and Integrate-and-Fire neurons, using spike-driven computation and second-order optimization to construct a multimodal spiking large language model, thereby achieving low-power and efficient inference.

**Unified Token Representations.** We project the history frames, the current frame, and the text into a shared latent space and concatenate them into a unified token sequence, which is then processed by the spiking large language model.

$$I_t = \begin{bmatrix} \mathbf{V}^h, \ \mathbf{V}^c, \ \mathbf{T} \end{bmatrix} \in \mathbb{R}^{(196 \times t + 196 + N_{\text{text}}) \times d}, \tag{6}$$

where $\mathbf{V}^h \in \mathbb{R}^{(196 \times t) \times d}$ are history visual tokens, $\mathbf{V}^c \in \mathbb{R}^{196 \times d}$ are current visual tokens, $\mathbf{T} \in \mathbb{R}^{N_{\text{text}} \times d}$ are text

tokens, $d$ is the shared hidden size, and $t$ is the number of history frames.

**Spiking Dynamics on Unified Tokens.** To convert unified token representations into sparse spike tokens, we evolve each token through spiking neuron dynamics and aggregate responses across fine-grained time steps. Each token receives a projected input and evolves with spike dynamics at fine time steps $t$:

$$
\begin{aligned}
\tilde{V}_{i,t} &= \alpha V_{i,t-1} + (I_{i,t} W_{\text{in}} + b) - \beta S_{i,t-1}, \\
S_{i,t} &= \mathbf{1}(\tilde{V}_{i,t} \geq V_{th}), \quad V_{i,t} = \tilde{V}_{i,t} - V_{th} S_{i,t},
\end{aligned}
\tag{7}
$$

where $I_{i,t}$ is the $i$-th token, $W_{\text{in}}, b$ are input projection parameters, $V_{i,t}$ is the membrane potential, $S_{i,t} \in \{0, 1\}$ is the spike, $\alpha \in (0, 1)$ is the leak, $\beta \geq 0$ models after the spike hyperpolarization, and $V_{th}$ is the threshold. To stabilize training, we merge $L$ consecutive fine-grained steps into a multi-level spike token:

$$
s_i[t'] = \sum_{t=t'L}^{(t'+1)L-1} S_{i,t}, \quad t' = 0, \dots, T'-1, T' = \frac{T}{L}. \tag{8}
$$

where $L$ denotes the merge window, $T$ and $T'$ denote the numbers of fine-grained time steps before and after merging, respectively, and $s_i(t') \in \{0, \dots, L\}$ is the spike count.

**Differential Temporal Sparsity Allocation.** To evaluate channel importance both across modalities and within each modality, we propose a differential temporal sparsity allocation mechanism. Specifically, informative channels are assigned a longer spiking horizon, while less important ones are encoded with a single-step spike. For a single modality, the spike-based reconstruction of the $c$-th channel at layer $l$ is formulated as:

$$
h_c^l = \frac{V_{th}}{T_c} \sum_{t=1}^{T_c} p_c^l[t] + z^{l-1}, \tag{9}
$$

where $p_c^l[t] \in \{0, 1\}$ denotes the spiking activity of the channel $c$ at the fine-grained step $t$, $T_c \in \{1, T'\}$ is the number of spiking steps assigned to the channel $c$, and $V_{th}$ is a spiking threshold per-token. $z^{l-1} = \min(h^{(l-1)})$ serves as a per-token zero-point shift, which allows spike-rate encoding for activations with negative values.

Considering channel importance across different modalities, we extend the channel-wise computation to the multimodal setting as follows:

$$
\tilde{h}_c^l = \frac{V_{th}}{T'} \sum_{t=1}^{T'} p^l[t] \Big|_{p \in C_{\text{m}}} + \min(\tilde{h}_c^{l-1}), \tag{10}
$$

where $C_{\text{m}}$ is the set of all channels within the current modality (e.g., visual or text). We employ second-order optimization to optimize the model, allowing for more precise weight adjustments and improving overall performance.

The framework unifies visual and language representations into a spiking token stream using spike coding, driving event-based processing to achieve efficient computation and energy savings.

### 3.4. Spiking Neural Network for Action Policy

We propose an action policy network based on Spiking Neural Networks. The input is encoded into spikes via population coding (Tang et al., 2021) and processed by multiple fully connected SNN layers, where neuronal currents, membrane potentials, and spike events are updated iteratively over time. Finally, the spiking action embedding is decoded into a continuous action vector through a decoding module. During reinforcement learning, the network is trained using the Proximal Policy Optimization (PPO) algorithm to optimize its parameters.

**Encoding and Decoding Network.** To encode continuous observation inputs into discrete spike outputs, we adopt population encoding. This approach transforms continuous observations into sparse and robust spike events, improving the stability and noise robustness of quadruped locomotion control. The encoding is given by the following formula:

$$
\mathbf{A}_E(s) = \Phi_{\text{LoG}}(s; \boldsymbol{\mu}, \sigma), \tag{11}
$$

where $\mathbf{A}_E(s) \in \mathbb{R}^K$ denotes the population responses for the scalar input $s$, and $\boldsymbol{\mu} = \{\mu_k\}_{k=1}^K$ and $\sigma$ are trainable receptive-field parameters initialized to cover the range of $s$.

**Deterministic Spike Encoding.** The population response $\mathbf{A}_E(s)$ serves as the stimulus for an input neuron population, where spikes are deterministically generated by simulating soft-reset integrate-and-fire (IF) dynamics. For the $k$-th neuron, the membrane state and spike emission are updated as follows:

$$
m_k^t = m_k^{t-1} + \lambda A_{E,k}, \tag{12}
$$

$$
y_k^t = \mathbf{1}(m_k^t \geq \theta), \tag{13}
$$

$$
m_k^t \leftarrow m_k^t - (\theta - \varepsilon) y_k^t, \tag{14}
$$

where $A_{E,k}$ denotes the stimulation strength of neuron $k$, $m_k^t$ is the membrane state at time $t$, $y_k^t \in \{0, 1\}$ is the emitted spike, $\lambda$ is an input scaling factor, $\theta$ is the firing threshold, and $\varepsilon$ is a small constant controlling the soft reset.

In the decoding module, we accumulate the spike count for each action dimension over $T$ time steps and compute the firing rate $fr(i)$ as the mean number of spikes per time step. The final action $a_i$ for each output dimension is then generated by applying the learned decoding weight and bias. This can be expressed as:

$$
a_i = W_d^{(i)} \cdot \frac{1}{T} \sum_{t=1}^{T} m_K^t + b_d^{(i)} \tag{15}
$$

Where $a_i$ is the action for the $i$-th output dimension, $W_d^{(i)}$ is the decoding weight for the $i$-th action dimension. $m_K^t$ represents the spike from the $i$-th output neuron in layer $K$ at time step $t$.

**Reinforcement Learning Training.** We train Spike-A with PPO under an actor–critic framework. At timestep $t$, the actor maps the observation $s_t$ to a policy $\pi_\theta(\cdot \mid s_t) = \mathcal{N}(\mu_t, \sigma_t)$ and samples an action $a_t$, while the critic estimates the state value $V_\phi(s_t)$ to compute the advantage $\hat{A}_t$. PPO stabilizes learning by clipping the policy ratio $r_t(\theta)$ to avoid excessively large updates. We train the spiking action policy network using surrogate-gradient spatiotemporal backpropagation, accumulating gradients over discrete timesteps $t = 1, \ldots, T$ and propagating them end-to-end through the spiking backbone as well as the learnable population encoder and decoder. Decoder parameters are updated independently for output populations $i = 1, \ldots, M$ corresponding to action dimensions, and encoder parameters $[\mu^{(i)}, \sigma^{(i)}]$ are updated independently for input populations $i = 1, \ldots, N$. We perform periodic updates over a fixed temporal window of length $T$, resulting in stable policies for continuous control.

We optimize the policy by maximizing the clipped surrogate objective. It can be formulated as follows:

$$J(\theta) = \mathbb{E}_t\big[\min\big(r_t(\theta)\hat{A}_t,\, c\big(r_t(\theta), 1-\epsilon, 1+\epsilon\big)\hat{A}_t\big)\big], \quad (16)$$

where $r_t(\theta) = \pi_\theta(a_t \mid s_t)/\pi_{\theta_{\text{old}}}(a_t \mid s_t)$ denotes the probability ratio between the current and previous policies, and $\hat{A}_t$ is the advantage estimate. The clipping operator bounds the effective update when $r_t(\theta)$ deviates beyond $[1-\epsilon,\, 1+\epsilon]$, preventing overly large policy changes and improving optimization stability.

For continuous action spaces, the actor parameterizes the policy as a gaussian distribution with mean $\mu_t$ and standard deviation $\sigma_t$, from which the action $a_t$ is sampled, providing fine-grained continuous control.

# 4. Experiments

## 4.1. Experimental Setups

**Dataset & Metric.** We evaluate vision-and-language navigation on VLN-CE R2R val-unseen for unseen-scene generalization, VLN-CE RxR val-unseen for long-horizon instruction following, and VLN-CE-Isaac for end-to-end navigation under realistic dynamics and traversability constraints. We use a unified benchmark that jointly evaluates navigation and low-level control. We evaluated VLN-CE R2R/RxR and VLN-CE-Isaac using a unified set of metrics, including NE, OS, SR, SPL, and nDTW, which capture goal-reaching accuracy, feasibility, success rate, path efficiency and trajectory fidelity, respectively. For low-level locomotion, we quantify command tracking and safety using linear and angular velocity tracking errors and the collision rate.

## 4.2. Main Results

**Navigation Performance on VLN-CE Benchmarks.** We evaluate SpikeVLA on R2R-CE and RxR-CE to assess environmental generalization under strict energy constraints, and to test whether competitive navigation performance can be maintained in resource-constrained conditions.

On R2R-CE Val-Unseen, we evaluate SpikeVLA under a resource-constrained setting with RGB-only observations and no waypoint supervision. As shown in Table 1, SpikeVLA achieves competitive navigation performance and slightly exceeds the strongest RGB baseline, NaVILA (Cheng et al., 2025), on some metrics. SpikeVLA substantially reduces inference cost, lowering GPU memory usage from 16.1 GB to 6.2 GB and achieving an energy metric of $E=49.09J$, which is approximately 34% of NaVILA. A comparison with AO-Planner shows that, although AO-Planner (Chen et al., 2025) uses additional sensor inputs such as panoramic views and depth, SpikeVLA achieves stronger core navigation metrics under the stricter RGB-only constraint. This suggests that the improvements come from more efficient representation learning and decision inference, rather than richer sensory inputs.

On RxR-CE Val-Unseen, the task presents higher language variability and a more challenging instruction distribution. Therefore, we evaluated SpikeVLA on this benchmark, providing both navigation performance and energy consumption. The current results suggest that SpikeVLA achieves performance comparable to strong baselines (Zhang et al., 2024a; Cheng et al., 2025) under the same RGB-only, no-waypoint settings, while maintaining the energy-efficiency advantages observed in R2R-CE, as shown in Table 3.

Comparison of resource consumption and performance between ANN-based and SNN-based architectures, shown in Fig. 3. The left side shows a comparison of resource consumption between the different components of SpikeVLA and NaVILA (Cheng et al., 2025), while the right side shows the performance comparison. It can be seen that SpikeVLA achieves comparable performance while using nearly one-third of the energy and computational resources of NaVILA. In general, SpikeVLA balances performance and efficiency, offering competitive navigation while reducing resource consumption for energy-constrained platforms.

**VLA Navigation Performance in Simulation.** We evaluated SpikeVLA in the VLN-CE-Isaac simulator using the Unitree Go2 platform to assess its transferability to closed-loop embodied execution under realistic dynamics and sensor noise. As shown in Table 2, it achieves a strong and stable navigation performance, maintaining high metric scores compared to NaVILA (Cheng et al., 2025). These results

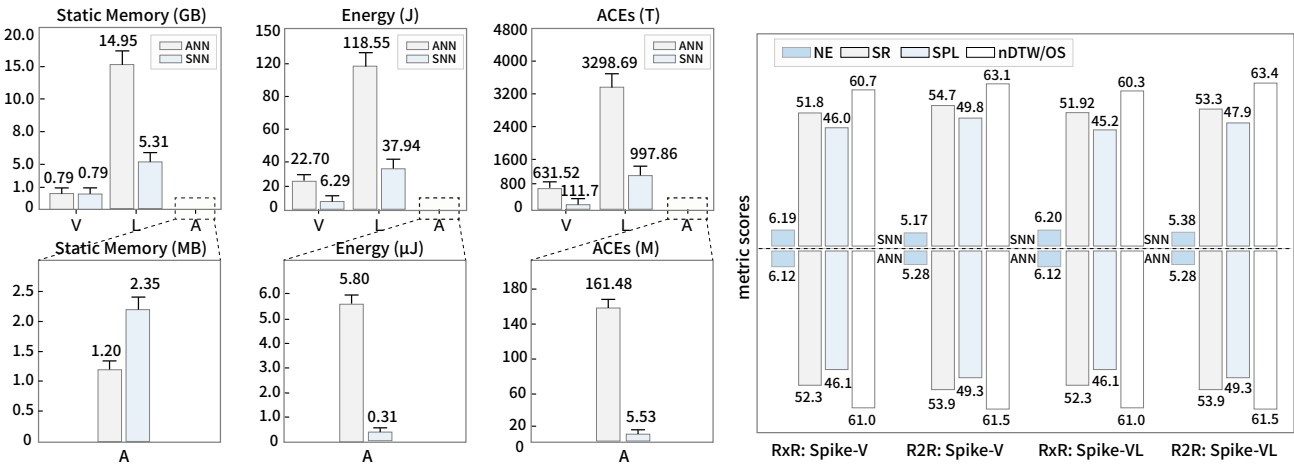

Figure 3: Comparison of resource consumption and performance between ANN-based and SNN-based architectures.

Table 1: **Comparison with SOTA methods on the VLN-CE Benchmarks.** The table summarizes navigation performance metrics NE, OS, SR, and SPL, together with resource-efficiency metric, where Mem denotes GPU memory usage and Eng denotes energy consumption. NaVid and UniNaVid have the same model parameters, resulting in equal energy consumption.

| Method | Observation | Waypoint | R2R Val-Unseen | | | | Resource Efficiency | | |
|---|---|---|---|---|---|---|---|---|---|
| | | | NE ↓ | OS ↑ | SR ↑ | SPL ↑ | Mem(MB) ↓ | Eng(J) ↓ | ACEs($10^{12}$) ↓ |
| CM2 (Georgakis et al., 2022) | RGB. & Depth & Odom. | ✗ | 7.02 | 41.0 | 34.0 | 27.0 | - | - | - |
| WS-MGMap (Chen et al., 2022) | RGB. & Depth & Odom. | ✗ | 6.28 | 47.0 | 38.0 | 34.0 | - | - | - |
| CMA (Hong et al., 2022a) | Pano. & Depth & Odom. | ✓ | 6.20 | 52.0 | 41.0 | 36.0 | - | - | - |
| Sim2Sim (Krantz & Lee, 2022) | Pano. & Depth & Odom. | ✓ | 6.07 | 52.0 | 43.0 | 36.0 | - | - | - |
| GridMM (Wang et al., 2023c) | Pano. & Depth & Odom. | ✓ | 5.11 | 61.0 | 49.0 | 41.0 | - | - | - |
| Ego$^2$-Map (Hong et al., 2023) | Pano. & Depth & Odom. | ✓ | 5.54 | 56.0 | 47.0 | 41.0 | - | - | - |
| DreamWalker (Wang et al., 2023a) | Pano. & Depth & Odom. | ✓ | 5.53 | 59.0 | 49.0 | 44.0 | - | - | - |
| HAMT+ScaleVLN (Wang et al., 2023b) | Pano. & Depth & Odom. | ✓ | 4.80 | - | 55.0 | 51.0 | - | - | - |
| ETPNav (An et al., 2024) | Pano. & Depth & Odom. | ✓ | 4.71 | 65.0 | 57.0 | 49.0 | - | - | - |
| HNR (Wang et al., 2024) | Pano. & Depth & Odom. | ✓ | 4.42 | 67.0 | 61.0 | 51.0 | - | - | - |
| AO-Planner (Chen et al., 2025) | Pano. & Depth | ✗ | 5.55 | 59.0 | 47.0 | 33.0 | - | - | - |
| NaVid (Zhang et al., 2024b) | RGB. | ✗ | 5.47 | 49.0 | 37.0 | 35.0 | 14231.96 | 157.29 | 4376.68 |
| UniNaVid(Zhang et al., 2024a) | RGB. | ✗ | 5.58 | 53.3 | 47.0 | 42.7 | 14231.96 | 157.29 | 4376.68 |
| NaVILA(Cheng et al., 2025) | RGB. | ✗ | 5.28 | 61.5 | 53.9 | 49.3 | 16119.98 | 141.25 | 3930.21 |
| MapNav(Zhang et al., 2025a) | RGB. | ✗ | 4.93 | 53.0 | 39.7 | 37.2 | - | - | - |
| **SpikeVLA(ours)** | RGB. | ✗ | 5.38 | 63.4 | 53.3 | 47.9 | 6249.18 | 49.09 | 1196.16 |

Table 2: **VLN-CE-Isaac evaluation results.** NaVILA result is reproduced under the same conditions. All experiments evaluated on 1,077 episodes in the VLN-CE-Isaac.

| | VLN-CE-Isaac | | | | Resource Efficiency | | |
|---|---|---|---|---|---|---|---|
| | NE ↓ | OS ↑ | SR ↑ | SPL ↑ | Mem(MB) ↓ | Eng(J) ↓ | ACEs($10^{12}$) ↓ |
| NaVILA-R | 6.29 | 52.1 | 36.5 | 29.5 | 16126.18 | 141.25 | 3930.21 |
| SpikeVLA | 6.02 | 53.6 | 32.7 | 28.5 | 6251.53 | 44.23 | 1109.61 |

indicate that the policy supports reliable task-oriented navigation behaviors when deployed in a closed-loop execution system. Overall, the simulation results validate the transition from offline benchmarks to embodied control, reinforcing SpikeVLA as a practical foundation for long-horizon deployment under energy constraints.

**Action Policy Performance.** We evaluate SpikeVLA

for low-level policy execution, focusing on control accuracy, safety, and energy efficiency. As shown in Table 4, SpikeVLA matches NaVILA (Cheng et al., 2025) overall, with only minor differences on a few metrics, while significantly reducing resource usage and energy consumption without sacrificing control quality. This advantage is primarily driven by the event-driven mechanism of spiking neural networks, which aligns more effectively with the deployment requirements of high-frequency closed-loop low-level control. We validated the control performance across various terrains, as shown in Fig 5.

**Comparison with Quantized VLA.** We further compared SpikeVLA with INT4-quantized VLA models, with the results summarized in Table 5. The results show that although INT4 quantization reduces computational and storage costs by lowering numerical precision, it leads to clear perfor-

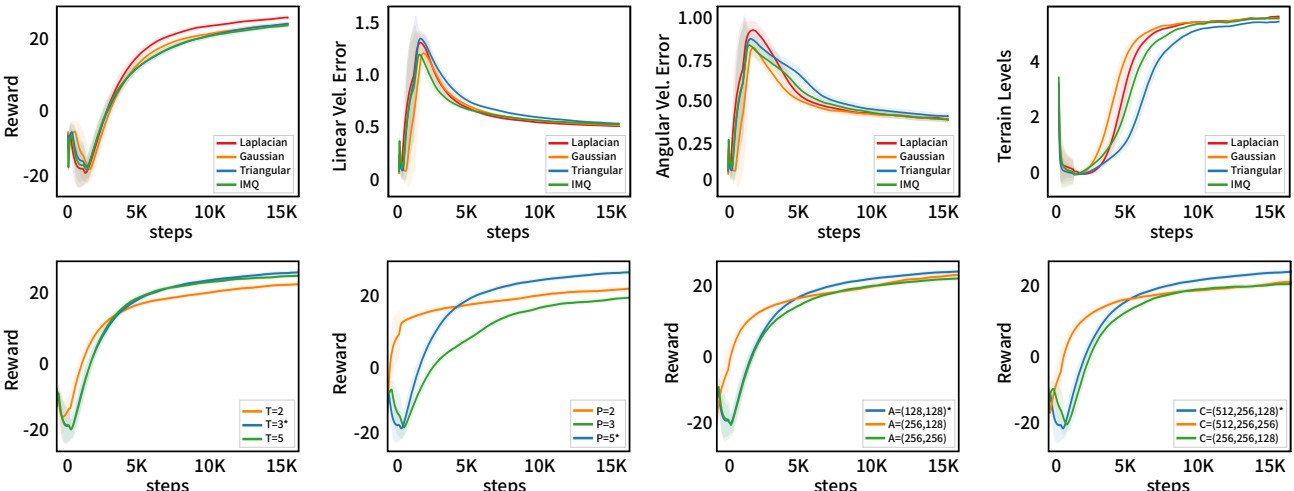

Figure 4: **SNN action policy network ablations.** The top row compares different spike encoding kernels (Laplacian, Gaussian, Triangular, and IMQ) in terms of reward, linear velocity error, angular velocity error, and terrain difficulty over training steps. The bottom row reports reward curves under different hyperparameter settings, including the time step $T$, population encoding size $P$, Actor network dimensions $A$, and Critic network dimensions $C$.

Table 3: **Comparison with SOTA methods on the Val-Unseen split of RxR-CE.** The table summarizes navigation performance metrics NE, SR, SPL and nDTW, together with resource-efficiency measures.

| Method | Observation | Waypoint | RxR Val-Unseen | | | | Resource Efficiency | | |
|---|---|---|---|---|---|---|---|---|---|
| | | | NE ↓ | SR ↑ | SPL ↑ | nDTW ↑ | Mem(MB) ↓ | Eng(J) ↓ | ACEs($10^{12}$) ↓ |
| BEVBert (An et al., 2022) | Pano. & Depth & Odom. | ✓ | 4.00 | 68.5 | - | 69.6 | - | - | - |
| CMA (Hong et al., 2022a) | Pano. & Depth & Odom. | ✓ | 8.76 | 26.5 | 22.1 | 47.0 | - | - | - |
| ETPNav (An et al., 2024) | Pano. & Depth & Odom. | ✓ | 5.64 | 54.7 | 44.8 | 61.9 | - | - | - |
| HNR (Wang et al., 2024) | Pano. & Depth & Odom. | ✓ | 5.50 | 56.3 | 46.7 | 63.5 | - | - | - |
| AO-Planner (Chen et al., 2025) | Pano. & Depth | ✗ | 7.06 | 43.3 | 30.5 | 50.1 | - | - | - |
| UniNaVid(Zhang et al., 2024a) | RGB. | ✗ | 6.24 | 48.7 | 40.9 | | 14231.96 | 157.29 | 4376.68 |
| NaVILA(Cheng et al., 2025) | RGB. | ✗ | 6.12 | 52.3 | 46.1 | 61.0 | 16119.98 | 141.25 | 3930.21 |
| **SpikeVLA(ours)** | RGB. | ✗ | 6.20 | 51.9 | 45.3 | 60.4 | 6249.18 | 49.09 | 1196.16 |

Table 4: Closed-loop performance of the low-level policy.

| Method | Linear Vel. | Angular Vel. | Resource Efficiency | | |
|---|---|---|---|---|---|
| | Error ↓ | Error ↓ | Mem(MB) ↓ | Eng(μJ) ↓ | ACEs($10^6$) ↓ |
| NaVILA | 0.23 | 0.38 | 1.20 | 5.80 | 161.48 |
| SpikeVLA | 0.42 | 0.29 | 2.35 | 0.31 | 5.53 |

Table 5: Comparison with ANN quantized model.

| Method | R2R Val-Unseen | | | | Resource Efficiency | | |
|---|---|---|---|---|---|---|---|
| | NE ↓ | OS ↑ | SR ↑ | SPL ↑ | Mem(GB) ↓ | Eng(J) ↓ | ACEs($10^{12}$) ↓ |
| NaVILA (FP16) | 5.28 | 61.5 | 53.9 | 49.3 | 15.7 | 141.25 | 3930.21 |
| NaVILA (INT4) | 5.66 | 56.8 | 48.2 | 43.6 | 8.6 | 72.49 | 982.55 |
| **SpikeVLA** | 5.38 | 63.4 | 53.3 | 47.9 | 6.1 | 49.09 | 1196.16 |

mance degradation. In contrast, SpikeVLA effectively reduces computational and storage costs while maintaining high task accuracy, thereby demonstrating a better trade-off between performance and efficiency.

It is important to emphasize that SpikeVLA and quantized ANN models differ fundamentally in their underlying mechanisms. Quantization mainly reduces computational cost by lowering the numerical precision of parameters and activations, whereas SNNs rely on an event-driven spiking computation mechanism in which computation is triggered only when neurons are activated, thereby reducing redundant operations. Therefore, SpikeVLA does not simply trade accuracy for efficiency. instead, it achieves higher energy efficiency through a sparse, event-driven computational paradigm.

### 4.3. Ablation Study

**Effect of Different SNN Modules.** Fig. 3 presents an ablation study that progressively converts the VLA components into spiking neural networks, where Spike-V, Spike-L, and Spike-A denote the spike vision encoder, multimodal spiking large language model, and spiking neural network for

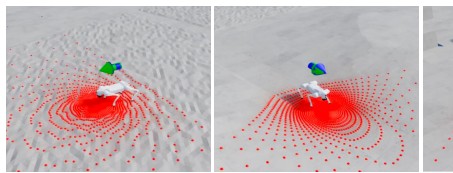

Figure 5: **Control performance across different terrain difficulties.** Left, middle, and right subfigures show rugged, sloped, and obstacle-filled terrains. The red points denote LiDAR point clouds, with green arrows representing the commanded velocity and blue arrows representing the actual velocity.

Table 6: Rewards across different population encoders. MEL donates Mean Episode Length, where higher values indicate better survival and task persistence.

| Kernel Classes | Rewards | MEL↑ | Resource Efficiency | | |
| --- | --- | --- | --- | --- | --- |
| | | | Mem(MB)↓ | Eng(μJ)↓ | ACEs($10^6$)↓ |
| ANN | 33.45 | 976.81 | 1.20 | 5.80 | 161.48 |
| Gaussian RBF kernel | 23.10 | 973.11 | 2.35 | 0.41 | 7.34 |
| Inverse Multiquadric kernel | 22.73 | 939.35 | 2.35 | 0.68 | 12.06 |
| Triangular kernel | 25.15 | 966.29 | 2.35 | 0.25 | 4.42 |
| Laplacian kernel | 26.72 | 983.94 | 2.35 | 0.31 | 5.53 |

action Policy, respectively. As spiking conversion is applied to additional components, navigation performance shows only a modest degradation, while compute and energy costs decrease monotonically with consistent additive benefits. The vision and reasoning modules play a key role in significantly reducing energy consumption. Extending the spiking control module brings additional efficiency improvements.

**Effect of Different Spike Encoding Mechanisms.** We find that Laplacian-kernel spike encoding achieves better performance on legged-robot control tasks. It represents continuous observations with sparse and robust population responses, improving the stability and noise robustness of quadruped locomotion control. Compared to the Gaussian RBF, whose squared-distance decay results in highly localized activations, the Laplacian kernel's exponential $\ell_1$-distance decay preserves sensitivity to mid-range variations. Moreover, compared to the hard truncation of the triangular kernel and the heavy-tailed long-range influence of the IMQ kernel, the Laplacian kernel offers a better trade-off between coverage and suppression of distant interference. Because foot-ground contacts, impact events, and terrain perturbations introduce pronounced transient dynamics, a stable yet not excessively smooth encoding generates more consistent population-fire activity, which supports downstream policy optimization and enhances the final return. Fig. 4 presents a comparison of the performance of different encoding kernels and the effect of different parameters in the action policy network on reward.

# 5. Conclusion

We present SpikeVLA, an event-driven paradigm for robotic tasks that reduces inference computation and energy consumption while maintaining the ability to execute complex instructions. This system involves a trade-off between performance and efficiency. Experiments across multiple VLN benchmarks and closed-loop simulations demonstrate robust generalization and stable performance, alongside reduced energy consumption. Its effectiveness has been validated through deployment on GPU platforms. The new SpikeVLA paradigm has a revolutionary impact on resource-constrained platforms, such as micro-robots and deep space exploration robots, driving the application of low-power, high-performance computing in extreme environments.

**Limitation and Future Work.** While SpikeVLA has demonstrated strong performance in simulations and on GPU platforms, its energy efficiency and real-time performance on neuromorphic chips remain unverified. Further validation on neuromorphic hardware will be the next step in our work. Moreover, we will focus on exploring SpikeVLA applications on energy-constrained platforms, such as micro-robots and robots for deep space exploration.

# Acknowledgements

This work was supported by the Joint Funds of the National Natural Science Foundation of China under Grant U24B20162 and the New Generation Artificial Intelligence-National Science and Technology Major Project (No. 2025ZD0124205).

# Impact Statement

SpikeVLA integrates spiking neural networks with vision-language-action models, enabling energy-efficient, brain-inspired multi-modal reasoning and control. This approach can support safer and more robust autonomous agents, assistive robotics, and sustainable AI applications. Careful attention to fairness and bias is essential to ensure responsible deployment in human-facing, high-stakes contexts such as healthcare and education.

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

# A. Dataset & Metric Details

## A.1. Dataset.

To comprehensively evaluate our method for vision-and-language navigation, we conduct experiments on three benchmarks: R2R-CE, RxR-CE, and VLN-CE-Isaac. R2R-CE is a continuous-control VLN benchmark set in photorealistic, reconstructed indoor environments. We evaluate on the val-unseen split to assess generalization across unseen scenes. Under the same continuous-control framework, RxR-CE presents a more challenging long-horizon navigation benchmark, including room-to-room traversal. We evaluate on the val-unseen split to assess robustness with complex instructions and extended navigation horizons. VLN-CE-Isaac is a high-fidelity benchmark built on Isaac Sim, where robots navigate indoor environments, with end-to-end evaluation conducted under realistic dynamics and traversability constraints.

## A.2. Metric.

We use a unified evaluation set that includes navigation performance, low-level control stability, and energy consumption. For R2R-CE, RxR-CE, and VLN-CE-Isaac, we provide the standard VLN metrics as follows.

### A.2.1. VLN METRICS.

- **Navigation Error (NE ↓)**: the Euclidean distance between the agent's final position and the goal, reflecting goal-reaching accuracy.

- **Oracle Success Rate (OS ↑)**: whether the agent enters the success threshold at any point along the trajectory, indicating the feasibility of reaching the goal.

- **Success Rate (SR ↑)**: Whether the final stop position falls within the success threshold, assessing task completion.

- **Success-weighted Path Length (SPL ↑)**: A success-based measure of path efficiency that penalizes unnecessary deviations and repeated steps, indicating whether the agent reaches the goal efficiently.

- **Normalized Dynamic Time Warping (nDTW ↑)**: A trajectory-alignment metric that measures the similarity between the predicted and reference paths in terms of both spatial geometry and temporal alignment, showing trajectory fidelity.

### A.2.2. LOW-LEVEL CONTROL METRIC.

For low-level control, we evaluate performance using the following metrics to assess command tracking accuracy and safety.

- **Linear Velocity Error ↓**: the deviation between the executed and target linear velocities, reflecting command tracking accuracy and motion stability.

- **Angular Velocity Error ↓**: the deviation between the executed and target angular velocities, reflecting turning and heading control accuracy.

### A.2.3. THEORETICAL ENERGY CONSUMPTION EVALUATION

To ensure reproducible and fair efficiency comparisons across different implementations and hardware platforms, we use theoretical energy consumption as our energy metric. This metric estimates inference energy based on the model's computational scale and spike sparsity. Consistent with previous classic SNN works (Yin et al., 2021; Yao et al., 2023; Horowitz, 2014), we assume that all operations are implemented with 45nm technology (Yin et al., 2021), where EMAC = 4.6pJ and EAC = 0.9pJ. This standardized approach is used to evaluate the energy efficiency of SpikeVLA in comparison to ANN baselines. SpikeVLA is being integrated with neuromorphic hardware and a quadruped robot to assess its performance in low-power systems, as mentioned in the future work section.

We decompose SPIKEVLA into a series of layers or computational blocks. For the $l$-th layer, the synaptic operations are defined as follows:

$$\mathrm{SOPs}_l = r_l \cdot T \cdot \mathrm{FLOPs}_l, \tag{17}$$

where $r_l$ denotes the average firing rate of the input spike train to the layer, $T$ is the number of discrete simulation time steps, and $\mathrm{FLOPs}_l$ is the corresponding dense compute of the layer, dominated by MAC operations.

We account for two types of elementary operations, multiply-and-accumulate (MAC) and spike-based accumulation (AC). The total end-to-end inference energy of SpikeVLA is computed as follows:

$$E_{\text{SpikeVLA}} = E_{\text{MAC}} \cdot \text{FLOPs}_1 + E_{\text{AC}} \cdot \sum_{l=2}^{L} \text{SOPs}_l, \tag{18}$$

where $L$ is the number of layers (or blocks), and $\text{FLOPs}_1$ denotes the dense compute of the first layer. The remaining layers are computed using their synaptic operations. In practice, we estimate $\text{FLOPs}_l$ and $r_l$ for each layer in Spike-V, Spike-L, and Spike-A, and sum their energies to obtain the total inference energy.

For ANN baselines, inference is fully performed with dense floating-point operations. The theoretical energy is therefore given by the following formula:

$$E_{\text{ANN}} = E_{\text{MAC}} \cdot \text{FLOPs}_{\text{ANN}}, \tag{19}$$

where $\text{FLOPs}_{\text{ANN}}$ represents the total FLOPs of the ANN model during inference. This definition aligns with the SpikeVLA framework to ensure fair and reproducible comparisons.

### A.2.4. ACE METRIC.

ACE, the Arithmetic Computation Effort (Zhang et al., 2022b), is designed to reflect the computational cost of neural network inference on idealized hardware, specifically hardware implemented with CMOS technology. It is defined as:

$$ACE = \sum_{i \in I, j \in J} n_{i,j} \cdot i \cdot j, \tag{20}$$

where $n_{i,j}$ denotes the number of multiply-accumulate operations (MACs) between $i$-bit and $j$-bit numbers, and $I$ and $J$ are sets of bitwidths used in the inference.

## B. Details of Spiking Neural Network Visual Encoder

Table 7: Comparison of final-layer features between ANN and SNN visual encoders after ImageNet calibration.

| Time-step $T$ | MSE | IFR | AFR | Energy (J) |
|---|---|---|---|---|
| 2 | 2.4648 | 0.3456 | 0.2372 | 0.3091 |
| 4 | 1.4209 | 0.4744 | 0.3526 | 2.2458 |
| 6 | 0.6577 | 0.3950 | 0.3777 | 4.4890 |
| 8 | 0.3704 | 0.3176 | 0.3665 | 6.2860 |
| 10 | 0.2456 | 0.2723 | 0.3497 | 7.7378 |
| 12 | 0.1863 | 0.2384 | 0.3324 | 8.9972 |
| 14 | 0.1498 | 0.2190 | 0.3169 | 10.1200 |
| 16 | 0.1237 | 0.2072 | 0.3034 | 11.1500 |

We use SigLIP-V2 as the ViT backbone, maintaining its original architecture, which includes patch embedding, 27 transformer encoder blocks, and an attention pooling head. To support event-driven temporal inference and reduce dense computation, we apply spiking wrappers to the main dense operators.

Table 7 compares the final-layer features between SigLIPv2 and the SNN vision encoder after ImageNet calibration. As the time-step increases from $T=2$ to $T=16$, the SNN vision encoder shows a gradual decrease in mean squared error (MSE), suggesting that longer time-steps improve feature alignment. Meanwhile, the stability of the firing rate increases, leading to an increase in energy consumption.

Although the energy consumption of the SNN vision encoder increases with longer time-steps, it maintains a strong balance between accuracy and energy efficiency. Overall, it demonstrates robust performance in long-horizon inference tasks, making it particularly suitable for energy-constrained applications.

# C. Additional Experiments

## C.1. Ablation of Different Modules in SpikeVLA

Table 8: **Ablation study on the Val-Unseen split of R2R-CE and RxR-CE across different modules of the SpikeVLA.**

| | $T$ | R2R-CE Val Unseen | | | | RxR-CE Val Unseen | | | | Resource Efficiency | | |
|---|---|---|---|---|---|---|---|---|---|---|---|---|
| | | NE↓ | OS↑ | SR↑ | SPL↑ | NE↓ | SR↑ | SPL↑ | nDTW↑ | Mem(MB)↓ | Eng(J)↓ | ACEs($10^{12}$)↓ |
| Navila | - | 5.28 | 61.5 | 53.9 | 49.3 | 6.12 | 52.3 | 46.1 | 61.0 | 16119.98 | 141.25 | 3930.21 |
| SpikeVLA(w/o V) | - | 5.56 | 61.61 | 52.80 | 47.69 | 6.34 | 50.61 | 44.35 | 59.55 | 6248.23 | 60.64 | 1629.38 |
| SpikeVLA(w/o L) | 8 | 5.63 | 61.34 | 50.57 | 46.13 | 6.63 | 48.16 | 41.86 | 58.61 | 16125.93 | 124.84 | 3410.44 |
| | 16 | **5.17** | 63.13 | **54.70** | **49.85** | 6.19 | 51.86 | 46.02 | 60.76 | 16125.93 | 129.70 | 3496.99 |
| SpikeVLA | 8 | 5.59 | 63.30 | 51.39 | 45.88 | 6.52 | 47.83 | 41.83 | 58.34 | **6249.18** | **44.23** | **1109.61** |
| | 16 | 5.38 | **63.40** | 53.29 | 47.90 | 6.20 | 51.92 | 45.25 | 60.35 | 6249.18 | 49.09 | 1196.16 |

Table 8 presents the ablation study results of the SpikeVLA model on the R2R-CE Val Unseen and RxR-CE Val Unseen datasets. The results show that applying Spike-L results in a slight decrease in navigation performance, but significantly enhances energy efficiency, particularly in terms of memory and energy consumption. In contrast, Spike-V improves path efficiency and success rate, while also boosting computational efficiency.

Overall, through SNN-based visual encoding and SNN-based language inference, SpikeVLA achieves a favorable balance between high performance and low energy consumption. Compared to the ablated versions of the vision or language modules, the full SpikeVLA model demonstrates enhanced overall performance, highlighting its advantages in long-horizon navigation tasks.

## C.2. Ablation Study of the Action Policy Model

### C.2.1. ABLATION STUDY OF HYPERPARAMETER TIME STEP

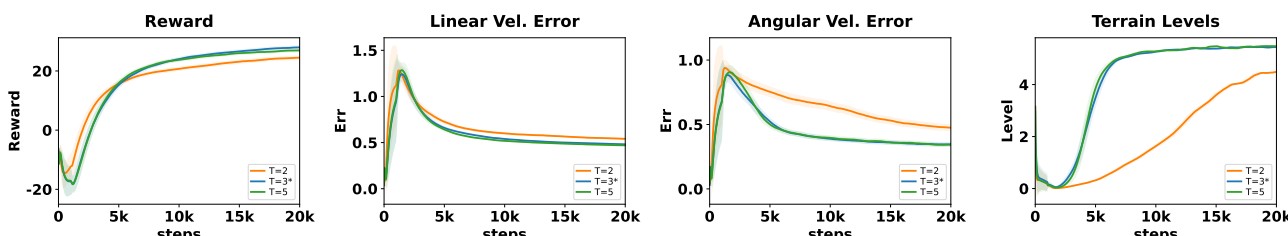

Figure 6: **Ablation study of the time step hyperparameter.** It shows the performance of different time step values (T = 2, 3, 5) across various metrics, including reward, linear velocity error, angular velocity error, and terrain levels.

Table 9: Ablation study of time step T of SNN action policy network. The baseline configuration is marked with $*$.

| Model | Configuration | | | | Tracking Error | | Resource Efficiency | | |
|---|---|---|---|---|---|---|---|---|---|
| | Time Step | Pop. Dim | Actor Dim. | Critic Dim. | Linear Vel.↓ | Angular Vel.↓ | Mem(MB)↓ | Eng(μJ)↓ | ACEs($10^6$)↓ |
| Spike-A | $T = 2$ | $P = 5$ | [128, 128] | [512, 256, 128] | 0.44 | 0.55 | 2.35 | 0.07 | 1.24 |
| | $T = 3^*$ | $P = 5$ | [128, 128] | [512, 256, 128] | 0.35 | 0.47 | 2.35 | 0.10 | 1.76 |
| | $T = 5$ | $P = 5$ | [128, 128] | [512, 256, 128] | 0.35 | 0.45 | 2.35 | 0.31 | 5.53 |

Table 9 presents the ablation study results of SpikeVLA-A across different timesteps $T$. As the time-step increases from $T$=2 to $T$=5, tracking accuracy improves, with the linear velocity tracking error decreasing from 0.44 to 0.35, and the angular velocity tracking error decreasing from 0.55 to 0.45, respectively. It suggests that longer time-steps improve the model's precision in dynamic tasks.

However, in terms of resource efficiency, memory consumption remains constant at $2.35MB$, while energy consumption and the ACE metric increase substantially with longer time-steps. Energy consumption increases from 0.07 $\mu J$ at T=2 to

0.31 $\mu J$ at T=5, while ACE rises from 1.24 to 5.53. It suggests that while longer time-steps improve accuracy, they also increase computational and energy costs, emphasizing the trade-off between accuracy and resource consumption when choosing the optimal time-step. Figure 6 demonstrates the effect of time step on the model's performance and stability across training steps.

### C.2.2. ABLATION STUDY OF HYPERPARAMETER POPULATION ENCODING SIZE

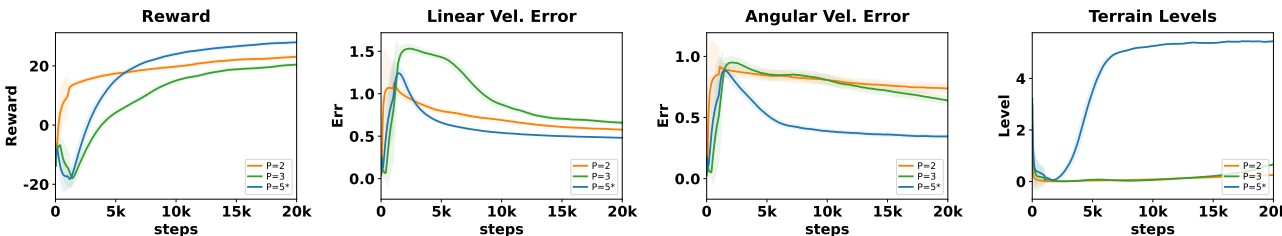

Figure 7: **Ablation study of the population encoding size hyperparameter.** It shows the performance of different population encoding dimensions (P = 2, 3, 5) across different metrics, including reward, linear velocity error, angular velocity error, and terrain levels.

Table 10: Ablation study of population encoding size P. The baseline configuration is marked with ∗.

| Model | Configuration | | | | Tracking Error | | Resource Efficiency | | |
|---|---|---|---|---|---|---|---|---|---|
| | Pop. Dim | Time Step | Actor Dim. | Critic Dim. | Linear Vel.↓ | Angular Vel.↓ | Mem(MB)↓ | Eng($\mu J$)↓ | ACEs($10^6$)↓ |
| | $P=2$ | $T=3$ | [128, 128] | [512, 256, 128] | 0.77 | 0.59 | 0.98 | 0.03 | 0.53 |
| Spike-A | $P=3$ | $T=3$ | [128, 128] | [512, 256, 128] | 0.65 | 0.68 | 1.43 | 0.15 | 2.62 |
| | $P=5^*$ | $T=3$ | [128, 128] | [512, 256, 128] | 0.35 | 0.47 | 2.35 | 0.10 | 1.76 |

Table 10 presents the ablation study results of SpikeVLA with different population encoding dimensions $P$. As the population encoding dimension increases from $P$=2 to $P$=5, tracking accuracy improves significantly, with the linear velocity tracking error decreasing from 0.77 to 0.35, and the angular velocity tracking error decreasing from 0.59 to 0.47. It suggests that increasing the population encoding dimension improves model accuracy, particularly in dynamic tasks, by giving for more precise feature representations.

However, in terms of resource efficiency, memory and energy consumption increase significantly as the population encoding dimension grows. Memory consumption increases from $0.98MB$ and energy consumption from $0.03\mu J$ at $P$=2, to $2.35MB$ and $0.10\mu J$ at $P$=5, while ACE increases from 0.53 to 1.76. This suggests that although increasing the population encoding dimension improves accuracy, it also results in higher computational and energy costs. Therefore, a balance between accuracy and resource consumption must be carefully considered in practical applications. Figure 7 demonstrates the effect of population encoding size on the model's performance and stability across training steps.

### C.2.3. ABLATION OF ACTOR NETWORK DIMENSIONS

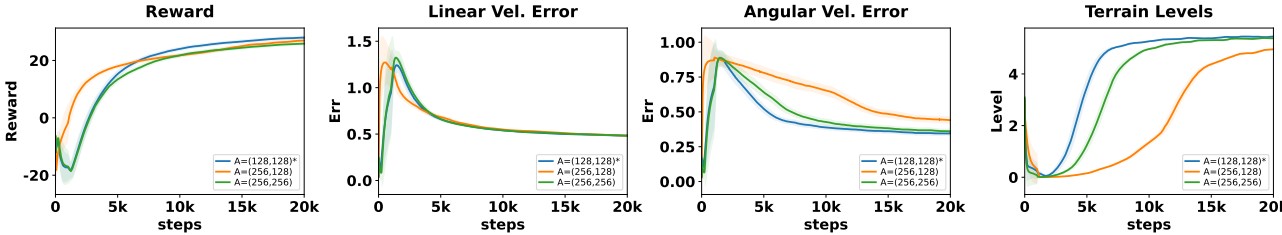

Figure 8: **Ablation of Actor Network Dimensions.** It shows the performance of different Actor network dimensions (A = [128, 128], A = [256, 128], A = [256, 256]) across various metrics, including reward, linear velocity error, angular velocity error, and terrain levels, evaluated over training steps.

Table 11 presents the ablation study results of SpikeVLA with different Actor network dimensions. As the Actor dimension increases from [128, 128] to [256, 128] and [256, 256], there is no significant improvement in accuracy, with a slight degradation in the angular velocity tracking error. The tracking error for linear velocity decreases from 0.35 to 0.34, while

Table 11: Ablation study of actor network dimensions. The baseline configuration is marked with ∗.

| Model | Configuration | | | | Tracking Error | | Resource Efficiency | | |
|---|---|---|---|---|---|---|---|---|---|
| | Actor Dim. | Time Step | Pop. Dim | Critic Dim. | Linear Vel.↓ | Angular Vel.↓ | Mem(MB)↓ | Eng(μJ)↓ | ACEs($10^6$)↓ |
| | [128, 128] | $T=3$ | $P=5$ | [512, 256, 128] | 0.35 | 0.47 | 2.35 | 0.10 | 1.76 |
| Spike-A | [256, 128]* | $T=3$ | $P=5$ | [512, 256, 128] | 0.43 | 0.48 | 4.63 | 0.17 | 2.96 |
| | [256, 256] | $T=3$ | $P=5$ | [512, 256, 128] | 0.34 | 0.49 | 4.48 | 0.34 | 6.14 |

the angular velocity error increases from 0.47 to 0.49, suggesting that increasing the dimension does not significantly improve accuracy and may even cause some performance degradation.

In terms of resource efficiency, memory consumption, energy consumption, and ACE increase significantly as the dimension grows. Memory consumption increases from 2.35MB and energy consumption from $0.10\mu J$ at [128, 128] to 4.48MB and $0.34\mu J$ at [256, 256]. This suggests that while larger Actor network dimensions may provide some accuracy improvements, they also incur higher computational and energy costs, emphasizing the trade-off between accuracy and resource consumption in practical applications. Figure 8 illustrates the impact of different Actor network dimensions on the model's performance across various metrics, including reward, linear velocity error, angular velocity error, and terrain levels.

### C.2.4. ABLATION OF CRITIC NETWORK DIMENSIONS

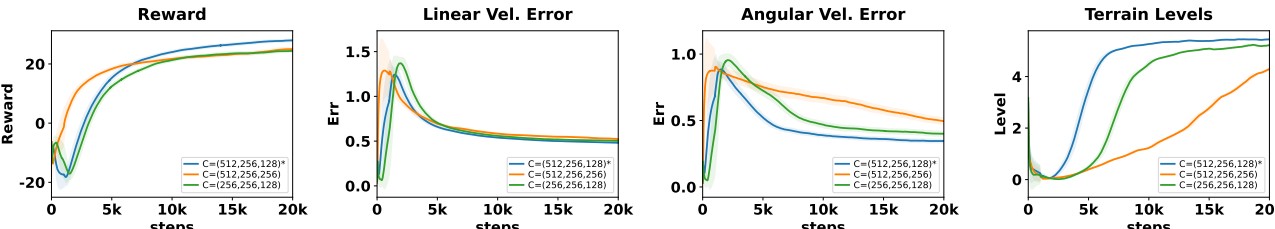

Figure 9: **Ablation of Critic Network Dimensions.** This figure shows the performance of different Critic network dimensions (C = [512, 256, 128], C = [512, 256, 256], C = [256, 256, 128]) across various metrics, including reward, linear velocity error, angular velocity error, and terrain levels, evaluated over training steps.

Table 12: Ablation study of critic network dimensions. The baseline configuration is marked with ∗.

| Model | Configuration | | | | Tracking Error | | Resource Efficiency | | |
|---|---|---|---|---|---|---|---|---|---|
| | Critic Dim. | Time Step | Pop. Dim | Actor Dim. | Linear Vel.↓ | Angular Vel.↓ | Mem(MB)↓ | Eng(μJ)↓ | ACEs($10^6$)↓ |
| | [512, 256, 128]* | $T=3$ | $P=5$ | [128, 128] | 0.35 | 0.47 | 2.35 | 0.10 | 1.76 |
| Spike-A | [512, 256, 256] | $T=3$ | $P=5$ | [128, 128] | 0.49 | 0.54 | 2.35 | 0.09 | 1.58 |
| | [256, 256, 128] | $T=3$ | $P=5$ | [128, 128] | 0.44 | 0.51 | 2.35 | 0.15 | 2.72 |

Table 12 presents the ablation study results of SpikeVLA with different Critic network dimensions. As the dimension increases from [256, 256, 128] to [512, 256, 128], accuracy improves, especially in the linear velocity tracking error, which decreases from 0.44 to 0.35, while the angular velocity error decreases from 0.51 to 0.47. This suggests that moderately increasing the Critic network dimensions improves accuracy.

However, when the Critic network dimensions are further increased to [512, 512, 256], accuracy drops significantly. The linear velocity error increases to 0.49, and the angular velocity error rises to 0.54. It indicates that excessively large network dimensions can degrade performance, likely due to model overfitting, which negatively impacts generalization and accuracy.

In terms of resource efficiency, memory consumption, energy consumption, and ACE values increase substantially with larger dimensions. Memory consumption increases from $2.35MB$ to $4.63MB$, energy consumption rises from $0.10\mu J$ to $0.15\mu J$, and ACE increases from 1.76 to 2.96. Overall, in practical applications, it is crucial to carefully balance accuracy and resource consumption when selecting the Critic network dimensions. Figure 9 illustrates the effect of different Critic network dimensions on the model's performance across various metrics, including reward, linear velocity error, angular velocity error, and terrain levels.

