# OpenReview forum: "SpikeVLA: Vision-Language-Action Models with Spiking Neural Networks"
_ICML.cc/2026/Conference — ICML 2026 regular_

### Official Review · Reviewer_1dWU · 2026-03-09

**Soundness:** 3
**Presentation:** 3
**Significance:** 3
**Originality:** 3
**Overall Recommendation:** 4
**Confidence:** 2

**Summary:**

Motivated by the high memory and energy cost of existing VLA model making the deployment on robotic platforms difficult, this work proposes to use a framework composed fully of SNN, instead of using SNN for a single perception or control module. To achieve this, this work proposes SpikeVLA, and it consists of three parts: a spiking visual encoder, a spiking multimodal language module, and a spiking action policy for embodied control. This design can preserve the core capabilities of VLA while leveraging sparse event-driven compution throughout the entire pipeline. The main contribution is its efficiency-performance trade-off, SpikeVLA remains competitive to ANN-based VLA methods on navigation tasks, but the memory and energy consumption is reduced.

**Compliance With Llm Reviewing Policy:**

Affirmed.

**Final Justification:**

The author has addressed my main concerns, and I have read other reviews and corresponding comments. I will maintain my current rating.

**Key Questions For Authors:**

Please refer to weakness.

**Limitations:**

yes

**Strengths And Weaknesses:**

**Strength:**
1. **The motivation is strong and design is novel.** This work aims clearly for the energy and memory constraint on robotic systems, I think this is an important and interesting direction for future development. The framework is entirely composed of SNN, and this is quite strong and interesting compared with replacing components with SNN.
2. **The performance is comparable while the resource savings are quite impressive.** Compared to NaVILA, SpikeVLA reduces memory and energy by around 3 times on R2R-CE and RxR-ce. while the performance is comparable. It's quite impressive.
3. **The first ablation study is quite convincing and complete.** From the experiment, it indeed reveals that adding more spiking components gives monotonic reductions in compute and energy with only modest performance degradation. And it aligns with the paper's main claim.

**Weakness:**
1. **The performance on neuromorphic chips remain unverified.** Since SpikeVLA's main selling point is the low resource consumption, but the real testing on a real-world device/platform is not conducted, which reduces the completeness of this work.
2. **The method's performance on RxR-CE and VLN-CE-Issac evaluation is not very competitive.** The proposed method's performance on R2R-CE is quite competitive, on RxR-CE, its performance is slightly lower than NaVILA, but on VLN-CE-Issace, the performance gap in SR and SPL is more noticeable.
3. **The second ablation study is not very convincing.** For the second ablation study which is to examine the effectiveness of different spike encoding mechanisms, the only advantage of Laplacian kernel is from the "Reward", and in later steps, from other metrices, all performance is quite close.

I'm willing to adjust my rating if my questions can be properly addressed.

---

> ### Author Rebuttal · Authors · 2026-03-30
>
> Thank you to the reviewer for the constructive comments. We appreciate the acknowledgment that "this is the first end-to-end Vision-Language-Action framework built entirely on Spiking Neural Networks" and the recognition of "impressive resource savings and convincing results of the ablation study."
>
> ---
>
> **Q1: The lack of real-world testing on neuromorphic chips.**
>
> We adopt the evaluation metrics consistent with previous classic SNN works [1,2,3], assuming that all operations are implemented with 45nm technology [1], where EMAC = 4.6pJ and EAC = 0.9pJ. SpikeVLA is being integrated with neuromorphic hardware and a quadruped robot to assess its performance in low-power systems, as mentioned in the future work section.
>
> Adapting neuromorphic chips to a quadruped robot platform requires system-level restructuring and co-optimization across multiple modules, not just a hardware replacement. Existing neuromorphic chips face challenges in operator support, compilation, deployment workflows, and compatibility with robot sensors and actuators. Additionally, achieving end-to-end closed-loop deployment under a spiking computation framework is a complex systems engineering problem.
>
> The current work focuses on validating the feasibility and efficiency of the SNN-VLA architecture under a unified evaluation framework, rather than full deployment on a specific neuromorphic platform.
>
> [1] Yin, Bojian, et al. Accurate and efficient time-domain classification with adaptive spiking recurrent neural networks. Nature Machine Intelligence (2021): 905-913.
>
> [2] Yao, Man, et al. Attention spiking neural networks. IEEE transactions on pattern analysis and machine intelligence (2023): 9393-9410.
>
> [3] Horowitz, Mark. 1.1 computing's energy problem (and what we can do about it). 2014 IEEE international solid-state circuits conference digest of technical papers (ISSCC). IEEE, 2014.
>
> ---
>
> **Q2: The proposed method's performance on R2R-CE is quite competitive, on RxR-CE, its performance is slightly lower than NaVILA, but on VLN-CE-Issace, the performance gap in SR and SPL is more noticeable.**
>
> On RxR-CE, SpikeVLA shows a slight performance drop compared to NaVILA, but with lower memory usage, energy consumption, and ACEs. On VLN-CE-Isaac, SpikeVLA outperforms NaVILA in NE and OS, with the main gap in SR and SPL, demonstrating retained navigation capability.
>
> This phenomenon is due to the representational characteristics of SNNs. Unlike ANN-VLA models, finite-step spikes introduce approximation errors, making semantic representations less sufficient in complex tasks with longer instructions. Since RxR-CE involves more complex scenarios and VLN-CE-Isaac involves handling complex joint tasks compared to R2R-CE, it leads to a certain degree of performance degradation. We view this as a trade-off between efficiency and representational capacity, rather than a fundamental limitation of the method itself.
>
> The efficiency advantage of SpikeVLA comes not just from model compression, but from the event-driven computation and sparse activation of SNNs, reducing redundant operations and resource consumption while maintaining task performance.
> We characterize the results as follows: SpikeVLA performs comparably to existing ANN methods across multiple benchmarks while offering significant resource-efficiency advantages. This is the core value of our work, and we will highlight this more clearly in the revised manuscript.
>
> ---
>
> **Q3: The Laplacian kernel only shows an advantage in "Reward," with other metrics showing similar performance.**
>
> We added Mean Episode Length (MEL) in Table 1, where higher values indicate better survival and task persistence. In experiments, SpikeVLA shows significant variation across encoding mechanisms, with the Laplacian kernel achieving the highest rewards and MEL.
>
> > Table 1: Comparison across different population encoders.
>
> | Kernel Classes              | Rewards | Mean Episode Length↑| Mem(MB)↓ | Eng(μJ)↓ | ACEs(10^6)↓ |
> | :-------------------------- | :-----: | :-----------------: | :------: | :------: | :---------: |
> | ANN                         |  33.45  |       976.81        |   1.20   |   5.80   |   161.48    |
> | Gaussian RBF kernel         |  23.10  |       973.11        |   2.35   |   0.41   |    7.34     |
> | Inverse Multiquadric kernel |  22.73  |       939.35        |   2.35   |   0.68   |    12.06    |
> | Triangular kernel           |  25.15  |       966.29        |   2.35   |   0.25   |    4.42     |
> | Laplacian kernel            |  26.72  |       983.94        |   2.35   |   0.31   |    5.53     |
>
> The results show that all spiking encoders outperform the ANN baseline in resource efficiency, with the Laplacian kernel achieving the highest reward, making it the best trade-off as the default.
> We will revise the manuscript to present the ablation conclusions more cautiously. For a detailed analysis, please refer to Tables 8–11 in the **Supplementary Material**.

---

> > ### Author Rebuttal · Reviewer_1dWU · 2026-04-02
> >
> > Thanks authors for your detailed response, my concerns have been addressed. And I'll keep my rating.

---

> > > ### Author Response · Authors · 2026-04-02
> > >
> > > Thank you for your thoughtful follow-up. We really appreciate it.
> > >
> > > We're glad to hear that our clarifications helped address your concerns. Thanks again for your thoughtful feedback and for taking the time to review our work.

---

### Official Review · Reviewer_coGK · 2026-03-11

**Soundness:** 3
**Presentation:** 2
**Significance:** 3
**Originality:** 3
**Overall Recommendation:** 4
**Confidence:** 4

**Summary:**

This paper aims to address the high inference cost and energy consumption of existing Vision-Language-Action (VLA) models, which limits their deployment on low-power real-time robotic platforms. The authors propose SpikeVLA, a fully spiking VLA framework consisting of three components: a spiking vision encoder (Spike-V), a spiking multimodal language model (Spike-L), and a spiking action policy network (Spike-A). The paper claims that, on VLN-CE navigation benchmarks and closed-loop control simulation, SpikeVLA achieves performance close to ANN-based baselines while substantially reducing GPU memory, energy consumption, and operator complexity. The overall positioning is clear: extend SNN / ANN-to-SNN conversion from isolated perception or control modules to the full VLA pipeline, emphasizing the efficiency benefits of event-driven computation.

**Compliance With Llm Reviewing Policy:**

Affirmed.

**Final Justification:**

This paper presents a novel attempt to reducing the inference latency and energy consumption of VLAs by introducing Spiking Neural Network (SNN), with a coherent three-stage design and solid experimental coverage across multiple tasks. The system-level integration of spiking components across perception, language, and control represents a notable contribution. The authors' responses well addressed my main concerns. I am happy to improve my original rating to a weak accept. I may also suggest the authors to polish the presentation of this paper.

**Key Questions For Authors:**

Please address the above weaknesses.

**Limitations:**

Yes

**Strengths And Weaknesses:**

Strengths:

1. The paper introduces SNNs into all three parts of the VLA stack—vision, language, and action—rather than replacing only a single encoder or controller. This system-level integration is a meaningful novelty. In particular, Spike-V applies spiking conversion to a SigLIP-style vision encoder, Spike-L attempts to construct a spiking multimodal language model over a unified token stream, and Spike-A uses population coding with an SNN-based policy for continuous control. The “three-stage” design is conceptually coherent and well motivated.

2. The paper provides extensive experimental results with multiple-level evaluations, including R2R-CE, RxR-CE, VLN-CE-Isaac, lower-level control experiments, and encoder ablations. Both task performance and resource efficiency are provided.

3. The target problem itself is meaningful and timely.

Weaknesses:

1. The energy evaluation is informative but limited in evidential strength:
The paper defines energy using a theoretical cost model based on FLOPs/SOPs and MAC/AC energy assumptions. This is useful as an efficiency-oriented analytical metric, but it is not equivalent to direct hardware-level power measurement. As a result, the paper supports the claim that SpikeVLA is more efficient under this theoretical model, but it does not fully validate real deployment-level energy savings on practical robotic systems. Some of the broader efficiency claims should therefore be phrased more carefully.

2. The empirical evidence supports a performance-efficiency trade-off more than a near-lossless replacement:
The appendix adds valuable ablations, and these are helpful. However, they also make it clearer that the gains come with non-negligible task-performance trade-offs in some settings. In particular, the evidence is more consistent with “meaningful efficiency gains at modest performance cost” than with stronger claims such as “without sacrificing control quality” or broadly “maintaining comparable performance” across tasks. The paper would be stronger if this trade-off were stated more explicitly and more cautiously.

3. The paper still lacks comparison to alternative efficiency strategies beyond SNN conversion:
The work is framed as an efficiency-oriented VLA approach, but the comparisons remain largely within the ANN-vs-SNN axis. The paper would be substantially stronger if it compared against other realistic efficiency baselines such as quantization, pruning, distillation, lightweight backbones, or other compute/memory reduction strategies. Without these comparisons, it is difficult to judge whether the proposed SNN formulation is the most compelling route to efficient VLA.

---

> ### Author Rebuttal · Authors · 2026-03-30
>
> We thank the reviewer for the careful comment and recognizing the novelty of introducing Spiking Neural Networks (SNNs) into all three components of the Vision-Language-Action (VLA) framework.
>
> ---
>
> **Q1: The energy evaluation, based on a theoretical model, supports SpikeVLA's efficiency but lacks validation through real-world hardware measurements. Some broader efficiency claims should be phrased more cautiously.**
>
> We agree that the energy evaluation in this work is based on a theoretical cost model using FLOPs/SOPs and MAC/AC assumptions, and should therefore be interpreted as a unified efficiency analysis rather than a direct hardware-level power measurement on a specific platform.
>
> At the same time, we would like to emphasize that this evaluation protocol is a widely adopted analytical paradigm in SNN research[1][2], especially when a unified neuromorphic hardware deployment setting is unavailable. Therefore, we believe the current results partially support the existing conclusion.
>
> We appreciate the reviewer’s suggestion and will revise the manuscript to more carefully distinguish between theoretical efficiency advantages and real deployment-level energy savings, and give more precisely claims.
>
> [1] Yin, Bojian, et al. Accurate and efficient time-domain classification with adaptive spiking recurrent neural networks. Nature Machine Intelligence(2021), 905-913.
>
> [2] Yao, Man, et al. Attention spiking neural networks. TPAMI(2023), 9393-9410.
>
> ---
>
> **Q2: More than a near-lossless replacement: The appendix adds valuable ablations, and these are helpful. The evidence is more consistent with “meaningful efficiency gains at modest performance cost” than with stronger claims such as “without sacrificing control quality” or broadly “maintaining comparable performance” across tasks. The paper would be stronger if this trade-off were stated more explicitly and more cautiously.**
>
> We thank the reviewer for the careful comment. We agree that the empirical results are more appropriately characterized as a performance efficiency trade-off. While SpikeVLA significantly reduces computation and energy consumption, it maintains competitive performance on most tasks, albeit with some performance degradation in certain settings.
>
> At the same time, we would like to emphasize that the observed performance changes are generally limited and should be interpreted in light of the substantial efficiency gains achieved. Therefore, we believe the core conclusion of this work is that SpikeVLA delivers significant efficiency improvements at only modest performance cost, rather than serving as a completely lossless replacement.
>
> We appreciate the reviewer’s suggestion and will revise the manuscript accordingly to moderate the relevant claims. Specifically, expressions such as “without sacrificing control quality” and broader statements such as “maintaining comparable performance” will be replaced with more careful and accurate wording, so as to better reflect the scope of the method and the actual experimental findings.
>
> ---
>
> **Q3: The paper lacks comparisons with other efficiency strategies such as quantization, pruning, and distillation.**
>
> We thank the reviewer for this important suggestion. We have added comparisons with quantized VLA models (INT4), and the results are summarized in Table 1. Preliminary results show that our method achieves better task performance while maintaining high energy efficiency. Although quantization methods achieve some energy efficiency improvements, they result in a significant decline in performance metrics.
>
> > Table 1: Comparison with ANN quantized model.
>
> | Method          | NE$\downarrow$ | OS$\uparrow$ | SR$\uparrow$ | SPL$\uparrow$ | Mem(GB)$\downarrow$ | Eng(J)$\downarrow$ | ACEs($10^{12}$)$\downarrow$ |
> | :-------------- | :------------: | :----------: | :----------: | :-----------: | :-----------------: | :----------------: | :-------------------------: |
> | NaVILA(Float16) |      5.28      |     61.5     |     53.9     |     49.3      |        15.7         |       141.25       |           3930.21           |
> | NaVILA (INT4)   |      5.66      |     56.8     |     48.2     |     43.6      |         8.6         |       72.49        |           982.55            |
> | SpikeVLA        |      5.38      |     63.4     |     53.3     |     47.9      |         6.1         |       49.09        |           1196.16           |
>
> SpikeVLA and quantized ANN models differ fundamentally. Quantization reduces cost by lowering precision, while SNNs use an event-driven mechanism that reduces redundant operations. SNNs also leverage temporal dynamics and sparsity, offering better energy savings in low-power scenarios, whereas quantized models rely on dense computation. Therefore, the advantage of SpikeVLA lies not only in numerical compression but more importantly in its computational mechanism, which offers greater potential for efficiency gains, especially on neuromorphic or sparsity-optimized hardware.

---

> > ### Author Rebuttal · Reviewer_coGK · 2026-04-04
> >
> > The rebuttal addressed my main concerns, especially by clarifying that the paper should be viewed as an efficiency–performance trade-off rather than a near-lossless replacement, and by adding the INT4 comparison. That said, the energy claims are still based on a theoretical model rather than hardware measurements, and the efficiency baselines are still not broad enough. I will raise my rating slightly.

---

> > > ### Author Response · Authors · 2026-04-04
> > >
> > > Thank you very much for your thoughtful feedback and hard work.
> > > We noticed that you mentioned planning to raise the rating, but it seems that the update hasn't been made yet. We were wondering if it might have been overlooked while updating the original reviewer comments.
> > > Again, we truly appreciate your recognition of our work and the time and effort you have devoted to the review.

---

### Official Review · Reviewer_LNpf · 2026-03-12

**Soundness:** 2
**Presentation:** 2
**Significance:** 3
**Originality:** 3
**Overall Recommendation:** 2
**Confidence:** 4

**Summary:**

This paper introduces SpikeVLA, an end-to-end SNN-based Vision-Language-Action architecture. It comprises Spike-V (vision encoder), Spike-L (multimodal LLM), and Spike-A (action policy) to replace dense Transformer computations with event-driven sparse activations, aiming to reduce energy consumption for resource-constrained robotics.

**Compliance With Llm Reviewing Policy:**

Affirmed.

**Final Justification:**

The author did not directly address the concerns raised in my office comments; furthermore, the Appendix was not integrated into a single PDF file with the main manuscript, thereby violating the submission guidelines. Consequently, I have lowered my rating.

**Key Questions For Authors:**

1. **Inference Speed:** The paper targets "latency-critical" tasks but omits actual latency metrics (e.g., FPS/ms). How does SpikeVLA compare quantitatively to baselines?
2. **Comparison with Quantized VLAs:** The paper demonstrates energy savings by comparing SpikeVLA to dense, full-precision ANN baselines. However, how does SpikeVLA compare (in both energy efficiency and task performance) against aggressively quantized (e.g., INT8/INT4) continuous VLA models? What is the fundamental or empirical advantage of this SNN architecture over highly quantized baselines on edge devices?
3. **RL Variance:** Can you provide the mean and standard deviation across 3-5 random seeds for the navigation and control metrics?
4. **Experimental Setup:** What is the exact number of time steps (T) used? Please provide the precise formulas and hardware constants used to calculate energy consumption.

**Limitations:**

**No.** While the authors mention that performance on physical neuromorphic chips remains unverified, they do not discuss qualitative failure modes. An analysis of specific vulnerabilities of the SNN architecture (e.g., semantic drift) compared to standard ANN VLAs could be included.

**Strengths And Weaknesses:**

**Strengths:**

- **Significance:** Addressing the high compute and energy bottlenecks in VLA models by shifting to an SNN paradigm is highly relevant.
- **Originality & Innovation:** This is the first work to propose an end-to-end Vision-Language-Action (VLA) framework built entirely on Spiking Neural Networks (SNNs). Shifting from dense, continuous Transformer computations to an event-driven paradigm is a highly novel approach to solving the compute and energy bottlenecks in embodied intelligence.

**Weaknesses:**

- **Missing Details:** The paper fails to define the time steps (T) used and the exact methodology/hardware assumptions for energy calculations (e.g., the 49.09 J metric).
- **Statistical Variance:** RL (PPO) is unstable, yet results lack multiple-seed testing and variance reporting.
- **Presentation:** The manuscript contains visible typos. For instance, the abstract misspells "consumption" as "onsumption" , and Figure 2 repeatedly misspells "linear" as "linner".

---

> ### Author Rebuttal · Authors · 2026-03-30
>
> We thank the reviewer for valuable comments and highlighting that "this is the first end-to-end VLA framework fully built on Spiking Neural Networks."
>
> ---
>
> **Q1: Definition of time steps (T) and the methodology/hardware assumptions for energy calculations.**
>
> T is the number of discrete time steps in the simulation process during a single forward pass of the SNN. Specifically, in the manuscript, T is set to 16 for Spike-V, 2 for Spike-L, and 5 for Spike-A. The choice of T is not empirical, and its impact is analyzed in **Tables 6–8 of our Supplementary Material,** including time-step ablations for all tasks.
>
> The reported 49.09 J is an energy estimate based on a unified theoretical model, following most previous SNN works [1, 2]. It assumes 45 nm technology [1], with EMAC = 4.6 pJ and EAC = 0.9 pJ. The energy calculation formulas are detailed in **Appendix A.2.3 and A.2.4**, with the theoretical basis cited in [3].
>
> [1] Yin, Bojian, et al. Accurate and efficient time-domain classification with adaptive spiking recurrent neural networks. Nature Machine Intelligence(2021), 905-913.
>
> [2] Yao, Man, et al. Attention spiking neural networks. TPAMI(2023), 9393-9410.
>
> [3] Horowitz, Mark. 1.1 computing's energy problem (and what we can do about it). ISSCC(2014).
>
> ---
>
> **Q2: Mean and standard deviation across random seeds for the navigation and control metrics.**
>
> In response, we conducted experiments with 3 random seeds, as shown in Table 1. All tasks demonstrate overall stability. We believe these variations do not impact the metrics and conclusions.
>
> > Table 1: Performance of navigation and control tasks across different seeds.
>
> |Method|Seed|Nav: NE↓|Nav: OS↑|Nav: SR↑|Nav: SPL↑|Control: Rewards↑|
> |-|-|-|-|-|-|-|
> |NaVILA|42|5.28|61.5|53.9|49.3|33.45|
> ||1|5.22|62.5|54.0|49.0|32.70|
> ||83|5.22|61.7|54.4|48.8|32.56|
> |SpikeVLA|42|5.38|63.4|53.3|47.9|26.72|
> ||1|5.48|62.0|52.8|47.7|25.95|
> ||83|5.30|62.6|53.2|48.1|27.26|
> |NaVILA|Mean ± Std|5.24±0.03|61.9±0.43|54.1±0.22|49.0±0.21|32.90±0.39|
> |SpikeVLA|Mean ± Std|5.36±0.08|62.7±0.57|53.1±0.22|47.9±0.16|26.64±0.54|
> ---
>
> **Q3: Comparison with Quantized VLAs.**
>
> We have added comparisons with quantized VLA models (INT4), and the results are summarized in Table 2. Although it improves energy efficiency, they lead to a significant decline in performance.
>
> > Table 2: Comparison with ANN quantized model.
>
> | Method          | NE↓ | OS↑ | SR↓ | SPL↑ | Mem(GB)↓ | Eng(J)$↓ | ACEs($10^{12}$)↓ |
> | :-------------- | :------------: | :----------: | :----------: | :-----------: | :-----------------: | :----------------: | :-------------------------: |
> | NaVILA(Float16) |      5.28      |     61.5     |     53.9     |     49.3      |        15.7         |       141.25       |           3930.21           |
> | NaVILA (INT4)   |      5.66      |     56.8     |     48.2     |     43.6      |         8.6         |       72.49        |           982.55            |
> | SpikeVLA        |      5.38      |     63.4     |     53.3     |     47.9      |         6.1         |       49.09        |           1196.16           |
>
> SpikeVLA and quantized ANN models are fundamentally different. Quantization reduces cost by lowering precision, while SNNs reduce redundant operations through an event-driven mechanism for better energy efficiency.
>
> ---
>
> **Q4: Comparison Inference Speed Baseline.**
>
> On mainstream GPUs, the sparsity of SNNs is not fully exploited. As noted in [4], although SNN activations are binary (0 or 1), current GPUs do not support binary acceleration. So, SNNs still rely on floating-point matrix multiplication, resulting in inference latencies comparable to those of ANNs. SNN fully leverages its speed and latency advantages only on neuromorphic chips. We will clarify this in the revised manuscript.
>
> [4] Li, Yuhang, et al. A free lunch from ANN: Towards efficient, accurate spiking neural networks calibration. PMLR(2021).
>
> ---
>
> **Q5: Qualitative failure mode discussion.**
>
> Compared with ANN-VLA models, one potential limitation of SpikeVLA is that its finite-step spiking representation may introduce a certain degree of approximation error. As a result, the scene with more complex visual semantics or long-horizon instruction constraints, the sufficiency of high-level semantic representation may be slightly affected. This should be viewed as a trade-off between efficiency and representational capacity, rather than a fundamental defect of the method itself.
>
> ---
>
> **Q6: Manuscript visible typos revise.**
>
> We apologize for the mistake. We will correct all misspellings and proofread the manuscript in the revised version.

---

> > ### Author Rebuttal · Reviewer_LNpf · 2026-04-02
> >
> > See my official comments.

---

> > > ### Author Response · Authors · 2026-04-03
> > >
> > > Thank you for your careful reading of our rebuttal and for keeping the discussion open. We deeply appreciate your constructive feedback.
> > >
> > > Due to the strict character limits during the initial rebuttal phase, we had to concisely merge several highly related weaknesses and questions to ensure we could address all your concerns within the allowed space. To clarify our response structure, here is the exact mapping of your original points to our rebuttal sections:
> > >
> > > * **Weakness 1 & Question 4 (Missing Details & Experimental Setup)** -> Addressed in **Rebuttal Q1** (Provided the definition and values of $T$, as well as the energy calculation details).
> > >
> > > * **Weakness 2 & Question 3 (Statistical & RL Variance)** -> Addressed in **Rebuttal Q2** (Provided 3-seed testing with mean/std, consolidating the results for **both navigation and control tasks** into Table 1).
> > >
> > >   We reiterate the consolidated mean and standard deviation (Mean ± Std) for both navigation and control tasks below.
> > >
> > >   **1. Navigation Tasks (3-Seed Mean ± Std)**
> > >
> > >   | Method                | NE ↓        | OS ↑        | SR ↑        | SPL ↑       |
> > >   | :-------------------- | :---------- | :---------- | :---------- | :---------- |
> > >   | **NaVILA** (Baseline) | 5.24 ± 0.03 | 61.9 ± 0.43 | 54.1 ± 0.22 | 49.0 ± 0.21 |
> > >   | **SpikeVLA** (Ours)   | 5.36 ± 0.08 | 62.7 ± 0.57 | 53.1 ± 0.22 | 47.9 ± 0.16 |
> > >
> > >   **2. Control Task (3-Seed Mean ± Std)**
> > >
> > >   | Method                | Rewards ↑    |
> > >   | :-------------------- | :----------- |
> > >   | **NaVILA** (Baseline) | 32.90 ± 0.39 |
> > >   | **SpikeVLA** (Ours)   | 26.64 ± 0.54 |
> > >
> > > * **Question 2 (Comparison with Quantized VLAs)** -> Addressed in **Rebuttal Q3** (Provided Table 2 with INT4 model comparisons).
> > >
> > > * **Limitation (Failure modes)** -> Addressed in **Rebuttal Q5** (Discussed approximation errors and semantic drift).
> > >
> > > *   **Weakness 3 (Typos & Presentation)** -> Addressed in **Rebuttal Q6**.
> > >
> > > ---
> > >
> > > **Supplement for Question 1 (Inference Speed)**,  we have added the exact quantitative metrics (FPS/ms) and provided the corresponding raw data. The exact quantitative latency metrics are evaluated on L20:
> > >
> > > *   **NaVILA (Baseline):** 1299.1ms/step
> > > *   **SpikeVLA (Ours):** 1915.6ms/step
> > >
> > > As discussed in Q4, current standard GPUs are optimized for dense matrix multiplications rather than the sparse, binary event-driven operations characteristic of SNNs. Because SpikeVLA simulates multiple time steps ($T$), its GPU latency is understandably longer than the ANN baseline. However, we emphasize that this is a hardware-specific limitation rather than an algorithmic flaw; despite the longer latency on standard GPUs, SpikeVLA significantly reduces theoretical energy consumption (as shown by our ACE metrics) and is highly promising for ultra-low power inference on dedicated neuromorphic hardware[1].
> > >
> > > **Supplement for Question 2**  & **Rebuttal Q3** (Provided Table 2 with INT4 model comparisons).  As shown in the table2, the metrics after INT4 quantization show a significant decline, while the SNN experimental results outperform the quantized model and offer advantages in energy consumption. Energy efficiency is realized when running on neuromorphic hardware.
> > >
> > > **Supplement for Question 4**, the hardware constants and precise details are provided in Rebuttal Q1 and the Appendix. To summarize the calculation framework, we decompose SpikeVLA into layers where the synaptic operations for the $l$-th layer are defined as:
> > >
> > > $SOPs_l = r_l \cdot T \cdot FLOPs_l$,
> > >
> > > with $r_l$ denoting the average firing rate, $T$ representing the discrete simulation time steps, and $FLOPs_l$ indicating the layer's dense compute. By accounting for both multiply-and-accumulate (MAC) and spike-based accumulation (AC) operations, the total end-to-end inference energy of SpikeVLA is computed as:
> > >
> > > $E_{SpikeVLA} = E_{MAC}\cdot FLOPs_1 + E_{AC}\cdot \sum_{l=2}^{L}SOPs_l$,
> > >
> > > where $FLOPs_1$ is the first layer's dense compute and $L$ is the total number of layers. To ensure fair and reproducible comparisons, the theoretical energy for ANN baselines, which rely entirely on dense floating-point operations, is analogously defined as:
> > >
> > > $E_{ANN}=E_{MAC}\cdot FLOPs_{ANN}$,
> > >
> > > where $FLOPs_{ANN}$ represents the total inference FLOPs.
> > >
> > > ---
> > >
> > > **Should there be any omissions or further questions, please feel free to point them out. We are more than happy to answer them promptly.**
> > >
> > > references:
> > >
> > > [1] Rajendran, Bipin, et al. Low-power neuromorphic hardware for signal processing applications: A review of architectural and system-level design approaches. IEEE Signal Processing Magazine 36.6 (2019): 97-110.

---

### Official Review · Reviewer_q1SG · 2026-03-13

**Soundness:** 3
**Presentation:** 3
**Significance:** 3
**Originality:** 4
**Overall Recommendation:** 5
**Confidence:** 2

**Summary:**

This work proposes the first Spiking Neural Network (SNN) - based VLA architecture for efficient energy consumption, particularly in low-resource embodied settings. Each of the three modules are converted to event-driven architectures referred to as Spike-V (vision), Spike-L (multimodal language + reasoning) and Spike-A (actor). The model is validated on three simulation benchmarks evaluating scene generalization, long-horizon modeling, and traversability constraints. SpikeVLA shows significant improvements over ANN-based VLAs in energy consumption and GPU memory use.

**Compliance With Llm Reviewing Policy:**

Affirmed.

**Final Justification:**

The authors conducted additional experiments and provided reasonable explanations to address questions I had about comparisons to prior work. I will keep my initial score.

**Key Questions For Authors:**

1. How would the authors compare Spike-V and Spike-A to the natively spiking variants of vision encoders and actors introduced in prior works?
2. A comparison of the proposed architecture against a spiking VLA obtained through ANN to SNN conversion is missing

**Limitations:**

Yes

**Strengths And Weaknesses:**

**Soundness** : The submission is technically sound, and the claims (reduced energy use while maintaining comparable performance to non-event-driven models) are supported by the simulation experiments and comparisons to ANN-based VLAs conducted, though
the model has not been evaluated yet on neuromorphic hardware. However, there are two questions unaddressed (please see below)

**Presentation** : The work is well motivated and for the most part, presented clearly. It would be useful to bold the State-of-the-Art numbers in the tables for quick reference. Also, Figure 5's caption can be made more self-contained.

**Significance** : This work will be of significant interest to the community, as it provides a new event-driven version for a broadly relevant class of architectures (VLAs).

**Originality** : The work is novel, since limited attention has been paid to making event-driven VLA architectures. The analysis of energy-consumption by different modules, particularly the vision and reasoning modules, might also be useful to identify targets for further energy optimization.

---

> ### Author Rebuttal · Authors · 2026-03-30
>
> We appreciate the reviewer’s valuable suggestion. Thank you to the reviewer for the positive feedback on the innovation, clear presentation, and significant impact of our work on the community.
>
> ---
>
> **Q1: The work is well motivated and for the most part, presented clearly. It would be useful to bold the SOTA numbers in the tables for quick reference. Also, Figure 5's caption can be made more self-contained.**
>
> Following the suggestions, we will revise the relevant tables by boldfacing the SOTA results. Further, we will refine the caption of Figure 5. Specifically, the left, middle, and right subfigures illustrate rugged terrain, sloped terrain, and obstacle-containing terrain, respectively. The red points denote LiDAR point clouds, the green arrows represent the commanded velocity, and the blue arrows represent the actual velocity.
>
> ---
>
> **Q2: How would the authors compare Spike-V and Spike-A to the natively spiking variants of vision encoders and actors introduced in prior works?**
>
> We believe that the main differences between Spike-V/A and existing related works lie in their design objectives and application scenes. For Spike-V, a direct comparison is not straightforward, as it is specifically designed for VLA tasks, where the vision encoder must support both high-level semantic understanding and action decision-making within a VLA closed loop. In contrast, most existing SNN vision encoders have been developed for foundational vision tasks such as classification and detection. The task objectives, input-output formats, and system roles differ from VLA models. A simple comparison may not be fair. Our focus is on validating whether a spiking vision encoder can effectively support VLA tasks while retaining efficiency advantages within the overall system.
>
> > Table 1: Comparison with another SNN Actor.
>
> | Methods     | Rewards↑ | Mem(MB)↓ | Eng(µJ)↓ | ACEs($10^6$)↓ |
> | :---------- | :-----: | :-----------------: | :-----------------: | :----------------------: |
> | NaVILA      |  33.45  |        1.20         |        5.80         |          161.48          |
> | ACSF-SNN[1] |  24.46  |        1.59         |        2.32         |          82.50           |
> | SpikeVLA    |  26.72  |        2.35         |        0.31         |           5.53           |
>
> For Spike-A, we compare our method with ACSF-SNN[1], which adopts a spiking action network, with the detailed results presented in Table 1. The results show that our method has certain advantages in the relevant metrics.
>
> [1] Lang Qin, et al. A low-latency adaptive coding spike framework for deep reinforcement learning. IJCAI (2023), 3049–3057.
>
> ---
>
> **Q3: A comparison of the proposed architecture against a spiking VLA obtained through ANN to SNN conversion is missing.**
>
> Conducting such a comparison at this stage remains challenging. Existing ANN-to-SNN conversion methods are mainly designed for basic vision tasks like classification and detection, while VLA integrates visual perception, language understanding, action decision-making, and closed-loop control, making it significantly different from conventional SNN models in both architecture and task objectives. Currently, no standard conversion pipeline is available for VLA.
> More importantly, our goal is not just to convert ANN-VLA into a spiking network, but to natively explore an event-driven design for VLA. To this end, we adopt the modular design of Spike-V, Spike-L, and Spike-A to validate the feasibility and efficiency potential of event-driven VLA architectures.
>
> > Table 2: Comparison with ANN quantized model.
>
> | Method          | NE↓ | OS↑ | SR↑| SPL↑| Mem(GB)↓| Eng(J)↓| ACEs($10^{12}$)↓|
> | :-------------- | :------------: | :----------: | :----------: | :-----------: | :-----------------: | :----------------: | :-------------------------: |
> | NaVILA(Float16) |      5.28      |     61.5     |     53.9     |     49.3      |        15.7         |       141.25       |           3930.21           |
> | NaVILA (INT4)   |      5.66      |     56.8     |     48.2     |     43.6      |         8.6         |       72.49        |           982.55            |
> | SpikeVLA        |      5.38      |     63.4     |     53.3     |     47.9      |         6.1         |       49.09        |           1196.16           |
>
> As an additional supplement, we include comparisons with ANN-quantized VLA models. The results are shown in Table 2, where INT4 quantization leads to a significant drop in navigation performance. In contrast, SpikeVLA achieves lower energy while maintaining performance close to ANN-VLA, demonstrating a better balance between performance and efficiency. Moreover, compared to quantization methods, SpikeVLA adopts an event-driven mechanism, where computation is triggered only by spikes, reducing redundant operations and exploiting activation sparsity, which provides greater potential for energy and computational efficiency.

---

> > ### Author Rebuttal · Reviewer_q1SG · 2026-04-03
> >
> > Thanks for the rebuttal response. Could you expand on why the Spike-V comparison is hard?
> > I agree that if trained on different tasks and different datasets, it would be an unfair comparison. However, as a spiking encoder on visual data, I'm asking to see if there's any core advantages in the representations produced by Spike-V. If the claim is that it's modular, it should be easily replaced by some other spiking encoder on your vision cues?
> >
> > Could the authors expand on what it would precisely take to either
> > (a) evaluate and compare Spike-V on classification/detection from video ?
> > (b) drop-in a different spiking vision encoder into your module

---

> > > ### Author Response · Authors · 2026-04-04
> > >
> > > We sincerely thank you for your thoughtful questions and insights, and appreciate the opportunity to clarify the role of Spike-V in our work. We agree that this is an important discussion and recognize the need to clearly define the requirements for evaluation and module replacement.
> > >
> > > ---
> > >
> > > **Q1: Why the Spike-V comparison is hard, and the nuances of its modularity**
> > >
> > > The difficulty in comparing Spike-V stems primarily from the fact that it was not trained as an independent vision backbone, but rather as an integral module within an end-to-end VLA system. To be compatible with the Spike-L for reasoning, Spike-V relies on cross-modal alignment (e.g., via image-text contrastive training) to effectively handle VLA tasks.
> > >
> > > Unlike unimodal vision models (such as ResNet or ViT), Spike-V must not only extract visual features but also align them with language representations. Consequently, its task objectives and application scenarios differ significantly from purely visual models. Current SOTA SNN vision models, such as SEW-ResNet[1] and SpikeFormer[2], serve primarily as foundational visual feature extractors and lack this inherent vision-language alignment capability. Therefore, the "modularity" of Spike-V does not imply that it can be trivially swapped with any other spiking encoder off-the-shelf.
> > >
> > > ---
> > >
> > > **Q2: What would it precisely take to evaluate and compare Spike-V on classification/detection?**
> > >
> > > To ensure a fair and interpretable comparison, the following conditions must be fulfilled:
> > >
> > > *   **Consistent Tasks and Datasets:** Spike-V and the baseline spiking encoders must be trained and tested on the exact same image classification or object detection benchmarks.
> > > *   **Adding task-specific heads and retraining:** To evaluate the performance, we should equip Spike-V with classification or bounding-box regression heads and retrain the model. The current checkpoint obtained from SpikeVLA can not be used for direct validation, as it has different learning objectives.
> > >
> > > To bridge this gap, we have supplemented our response with an **ImageNet-1K** classification experiment:
> > >
> > > Table 1 Classification task comparison experiment on ImageNet-1K
> > >
> > > | Model              | Top-1 Accuracy (%) |
> > > | :----------------- | :----------------- |
> > > | SEW-ResNet [1]     | 69.26              |
> > > | Spikingformer [2]  | 77.64              |
> > > | **Spike-V (Ours)** | **81.20**           |
> > >
> > > Table 1 shows the Top-1 accuracy of different models on a classification task. Spike-V achieves the highest accuracy of 81.2%, outperforming both SEW-ResNet[1] and Spikingformer[2], highlighting its feature extraction capability.
> > >
> > > ---
> > >
> > > **Q3: What would it precisely take to drop in a different spiking vision encoder into your module?**
> > >
> > > In our system, the vision encoder is architecturally separable and replaceable. However, a practical "drop-in" replacement still requires the following steps:
> > >
> > > *   **Vision-Language Cross-Modal Alignment Training**: In SpikeVLA, visual features need to be aligned with the language tokens. When replacing the vision encoder, the new SNN model must go through vision-language alignment training to ensure that its extracted features can be effectively used by Spike-L for reasoning.
> > > *   **Feature Interface Compatibility:** The new encoder must produce visual features compatible with the original Spike-V in both spatio-temporal structure and feature dimensions. This usually requires adding projection or adapter layers to align the features, which typically involves fine-tuning.
> > >
> > > Most SNN research currently focuses on basic vision tasks. Therefore, Spike-V's "modularity" means it is structurally separable, but cannot be easily replaced by any spiking vision encoder without adaptation and retraining. SEW-ResNet [1] and SpikeFormer[2] lack vision-language multimodal alignment, and direct replacement would greatly reduce the navigation capability of the VLA model.
> > >
> > > ---
> > >
> > > **Summary**
> > >
> > > We thank the reviewer for encouraging us to reconsider the main focus of this paper. Substituting the encoder with a more powerful spiking vision model would enhance overall system performance, which is a promising direction for future research.  We greatly appreciate the reviewer for highlighting this potential path, and we will clearly define the scope of our modular design in the revised manuscript.
> > >
> > > Thank you again for your thoughtful suggestions and helpful feedback, which have been invaluable in improving our paper.
> > >
> > > ---
> > >
> > > References:
> > >
> > > [1] Fang, Wei, et al. Deep residual learning in spiking neural networks. Advances in neural information processing systems 34 (2021): 21056-21069.
> > >
> > > [2] Zhou, Chenlin, et al. Spikingformer: A key foundation model for spiking neural networks. Proceedings of the AAAI Conference on Artificial Intelligence. Vol. 40. No. 3. 2026.

---

### Decision · Program_Chairs · 2026-04-30

**Decision:**

Accept (regular)

**Comment:**

This paper proposes an end-to-end spiking VLA framework for VLN, targeting better efficiency with competitive performance. Overall, the reviews are positive. `q1SG` finds the method technically sound and keeps a clear accept after rebuttal, noting that the added experiments address prior comparison concerns. `1dWU` considers the concerns fully resolved and keeps the rating. `coGK` also finds the main issues addressed and indicates a slight score increase, especially appreciating the added INT4 baseline and the clarified efficiency–performance positioning. Overall, reviewers agree the problem is important and the approach is novel.

The main remaining concern comes from `LNpf`. The reviewer points to gaps in the experimental validation, including limited details on energy estimation, missing latency analysis, and insufficient comparison to efficiency baselines. There is also concern that the energy claims rely on a model rather than hardware measurement, and that the rebuttal does not fully address the follow-up questions. These issues weaken the evidence but do not outweigh the positive feedback from the other reviewers. I therefore recommend a weak accept.